# Better SGD using Second-order Momentum

**Hoang Tran**
Boston University
tranhp@bu.edu

**Ashok Cutkosky**
Boston University
ashok@cutkosky.com

## Abstract

We develop a new algorithm for non-convex stochastic optimization that finds an $\epsilon$-critical point in the optimal $O(\epsilon^{-3})$ stochastic gradient and Hessian-vector product computations. Our algorithm uses Hessian-vector products to "correct" a bias term in the momentum of SGD with momentum. This leads to better gradient estimates in a manner analogous to variance reduction methods. In contrast to prior work, we do not require excessively large batch sizes, and are able to provide an adaptive algorithm whose convergence rate automatically improves with decreasing variance in the gradient estimates. We validate our results on a variety of large-scale deep learning architectures and benchmarks tasks.

## 1 Introduction

In the last couple of decades, first-order algorithms such as Stochastic Gradient Descent (SGD) or Adam [Kingma and Ba, 2014] have emerged as the main workhorse for modern Machine Learning (ML) tasks. They achieve good empirical results while being easy to implement and requiring relatively few computational resources. When the objective function is convex, SGD's convergence is well-understood [Zinkevich, 2003]. However, the results are much less favorable for the non-convex setting in which modern deep learning models operate. In fact, the problem of optimizing non-convex function is NP-hard in general, so instead analysis often focuses on finding a critical point - that is, a point at which the gradient of the loss is zero. SGD is well-known to find an $\epsilon$-approximate critical point in at most $O(\epsilon^{-4})$ total stochastic gradient evaluations [Ghadimi and Lan, 2013]. In an effort to improve upon SGD, many different algorithms and heuristics have been proposed, including various advanced learning rate schedules [Loshchilov and Hutter, 2016, Goyal et al., 2017, Xie et al., 2020], or per-coordinate learning rates and adaptive algorithms [Duchi et al., 2011, McMahan and Streeter, 2010, Kingma and Ba, 2014, Reddi et al., 2018]. All of these methods have enjoyed practical success, but none of their convergence rates has shown any asymptotic benefit over the $O(\epsilon^{-4})$ rate of SGD, which is to be expected because this rate is in fact *optimal* in the worst-case for first-order methods operating on smooth losses [Arjevani et al., 2019]. Thus, in order to make improved algorithms, we need additional assumptions. In this work, we consider the case in which our algorithm is not a pure first-order method, but has some limited access to second-order information.

Second-order algorithms such as Newton's method are among the most powerful methods in optimization theory. Instead of using a linear approximation for the objective via the gradient, Newton's method employs a quadratic approximation using both the gradient and the Hessian. This quadratic approximation hugs the curvature of the error surface and allows each iteration to make much faster progress. Unfortunately, second-order algorithms are also much more complex and expensive to implement. The cost of forming the Hessian matrix is $O(d^2)$ and the cost of a Newton step is typically $O(d^3)$, thus rendering such algorithms largely impractical for large-scale learning problems [Bottou and Bousquet, 2007].

The good news is we may not have to explicitly compute the Hessian matrix to be able to take advantages of the nice properties of second-order methods. In particular, it is possible to compute a *Hessian-vector product*, that is a vector of the form $\nabla^2 F(x)v$ for arbitrary $x$ and $v$, in roughly

36th Conference on Neural Information Processing Systems (NeurIPS 2022).

the same time it takes to evaluate $\nabla F(x)$ [Pearlmutter, 1994]. In this paper, we develop a novel second-order SGD-based algorithm that uses a stochastic gradient and Hessian-vector product oracle to expedite the training process. Our algorithm not only enjoys optimal theoretical properties, it is also practically effective, as demonstrated through our experimental results across various deep learning tasks. Importantly, our method works well on a *variety* of tasks in different domains, while standard practice is to employ different optimizers for different domains (e.g. AdamW for NLP, SGD for computer vision).

## 1.1 Contributions

We present a novel algorithm based on SGD with momentum (SGDHess) that uses Hessian-vector products to "correct" a bias term in the momentum. Through our theoretical analysis, we show that our algorithm requires the optimal $O(\epsilon^{-3})$ oracle calls for finding stationary points [Arjevani et al., 2020b]. In contrast to previous algorithms with this property (e.g. [Arjevani et al., 2020b]), we do not require excessively large batch sizes, and we also feel that our algorithm and analysis is relatively more straightforward: our method is a simple modification to SGD that could easily be applied to any momentum-based optimizer.

We also provide a variant of our algorithm based on normalized SGD, which dispenses with a Lipschitz assumption on the objective, and another variant with an *adaptive* learning rate that automatically improves to a rate of $O(\epsilon^{-2})$ when the noise in the gradients is negligible.

Finally, we test our algorithm on multiple learning tasks on different deep architectures. In all of these tasks, our algorithm consistently matches or exceeds the best algorithms for the task. Further, the tuning process of our algorithm is reasonably simple; in many cases, we can use the exact parameters of SGD for the new algorithm.

## 1.2 Related works

Although second-order methods have many attractive theoretical properties, making these methods practical is challenging. There have been multiple efforts to develop efficient second-order algorithms. One popular approach is the Broyden–Fletcher–Goldfarb–Shanno algorithm (BFGS) [Keskar and Wächter, 2019, Liu and Nocedal, 1989, Bollapragada et al., 2018, Pan et al., 2017], which provides a faster approximation to the Newton step using only a first-order oracle. Although BFGS and similar methods have been applied with some success to deep learning tasks [Bollapragada et al., 2018, Ma, 2020], we stress that such methods are *fundamentally incapable* of matching the convergence guarantees available to algorithms that truly use the Hessian in the stochastic non-convex setting [Arjevani et al., 2019].

More recent attention has focused on the use of Hessian-vector products as a way to achieve some of the benefits of second-order information without incurring significant computational overhead [Carmon et al., 2018, Agarwal et al., 2017]. [Tripuraneni et al., 2017] uses Hessian-vector products to approximate the cubic regularized Newton method [Nesterov and Polyak, 2006], converging in only $O(\epsilon^{-3.5})$ stochastic oracle calls. [Zhou et al., 2019] leverages both Hessian information and variance-reduction techniques to achieve a convergence rate of $O(\frac{n^{4/5}}{\epsilon^{3/2}})$ for finite-sum problems with $n$ summands. However, both algorithms fail to achieve the optimal $O(\epsilon^{-3})$ rate and to our knowledge none of these have been tested extensively on deep learning benchmarks. The first algorithm achieving the optimal $O(\epsilon^{-3})$ using Hessian-vector products rate was introduced in [Arjevani et al., 2020a]. However, their algorithm requires the batch size to be roughly $O(T^{1/3})$, which is excessively large for practical use-cases. On the other hand, our algorithm will work with *any* batch size.

Finally, variance-reduction based algorithms [Johnson and Zhang, 2013] can also achieve this same $O(T^{1/3})$ rate for non-convex optimization [Fang et al., 2018, Zhou et al., 2018, Cutkosky and Orabona, 2019, Tran-Dinh et al., 2019]. Despite favorable theoretical guarantees, variance-reduction algorithms often rely on stronger smoothness assumptions, are tricky to implement properly, and seem to not perform as well as expected in practice [Defazio and Bottou, 2018]. Comparison between variance-reduction algorithms and our algorithm (SGDHess) will be further discussed at the end of the paper.

## 1.3 Problem setup

We are interested in minimizing a function $F$ given by:

$$F(\vec{x}) = \mathop{\mathbb{E}}_{z \sim P_z} [f(\vec{x}, z)]$$

Where $f(\vec{x}, z)$ is a differentiable function of $\vec{x}$. $\vec{x}$ indicates the model parameters (e.g. the weights of some neural networks), $z$ indicates an example data point[1]. We are also given a point $\vec{x}_1$ and define $\Delta \in \mathbb{R}$ by:

$$\sup_{\vec{x} \in \mathbb{R}^d} F(\vec{x}_1) - F(\vec{x}) = \Delta \tag{1}$$

Further, we assume $F$ is $L$-smooth. That is, for all $\vec{x}$ and $\vec{y}$:

$$\|\nabla F(\vec{x}) - \nabla F(\vec{y})\| \le L\|\vec{x} - \vec{y}\| \tag{2}$$

To design a second-order algorithm, we also assume access to second-order stochastic oracle. Given any $\vec{x}$ and vector $\vec{w}$, we are allowed to sample $z \sim P_z$ and compute $\nabla f(\vec{x}, z)$ and $\nabla^2 f(\vec{x}, z)\vec{w}$. We assume that for all $\vec{x}$ and $\vec{w}$:

$$\mathbb{E}[\|\nabla f(\vec{x}, z) - \nabla F(\vec{x})\|^2] \le \sigma_G^2 \tag{3}$$

$$\mathbb{E}[\|\nabla^2 f(\vec{x}, z)\vec{w} - \nabla^2 F(\vec{x})\vec{w}\|^2] \le \sigma_H^2\|\vec{w}\|^2 \tag{4}$$

In addition, we will also need the assumption of second-order smoothness. For all $\vec{x}$, $\vec{y}$, and $\vec{w}$:

$$\|(\nabla^2 F(\vec{x}) - \nabla^2 F(\vec{y}))\vec{w}\| \le \rho\|\vec{x} - \vec{y}\|\|\vec{w}\| \tag{5}$$

Finally, in two of our algorithms, we need to assume $f$ is $G-$Lipschitz.

$$\|\nabla f(\vec{x}, z)\| \le G \tag{6}$$

## 2 SGD with Hessian-corrected momentum

In this section, we provide our SGD-based second-order algorithm. Before we go into the analysis, let us compare our algorithm to SGD to see why adding the Hessian-vector product term to the momentum update is a good idea. The standard SGD with momentum update is the following:

$$\hat{g}_t = (1 - \alpha)\hat{g}_{t-1} + \alpha\nabla f(\vec{x}_t, z_t)$$
$$\vec{x}_{t+1} = \vec{x}_t - \eta\hat{g}_t$$

To gain some intuition for this update, let us suppose that $\hat{g}_{t-1} = \nabla F(\vec{x}_{t-1})$. Then, viewing $\hat{g}_t$ as an estimate of $\nabla F(\vec{x}_t)$, define the error as:

$$\hat{\epsilon}_t = \hat{g}_t - \nabla F(\vec{x}_t)$$

Intuitively, SGD will converge rapidly if this error is small. We can write:

$$\hat{\epsilon}_t = (1 - \alpha)(\hat{g}_{t-1} - \nabla F(\vec{x}_t)) + \alpha(\nabla f(\vec{x}_t, z_t) - \nabla F(\vec{x}_t))$$

The second term is fairly benign: it is zero in expectation and is multiplied by a potentially small value $\alpha$. However, the first term is a bit trickier to bound since $\nabla F(\vec{x}_{t-1}) \ne \nabla F(\vec{x}_t)$. Our approach is to leverage the second-order oracle to improve this estimate. Specifically, we modify the momentum update to:

$$\hat{g}_t = (1 - \alpha)(\hat{g}_{t-1} + \nabla^2 f(\vec{x}_t, z_t)(\vec{x}_t - \vec{x}_{t-1})) + \alpha\nabla f(\vec{x}_t, z_t)$$

Now, if $\hat{g}_{t-1} = \nabla F(\vec{x}_{t-1})$, we would have

$$\mathbb{E}[\hat{g}_{t-1} + \nabla^2 f(\vec{x}_t, z_t)(\vec{x}_t - \vec{x}_{t-1})] = \hat{g}_{t-1} + \nabla^2 F(\vec{x}_t, z_t)(\vec{x}_t - \vec{x}_{t-1})$$
$$\approx \nabla F(\vec{x}_t) + O(\|\vec{x}_t - \vec{x}_{t-1}\|^2)$$

---

[1] $z$ could also indicate a minibatch of examples

Further, we have:

$$\hat{\epsilon}_t = (1 - \alpha)(\hat{g}_{t-1} - \nabla F(\vec{x}_t) + \nabla^2 f(\vec{x}_t, z_t)(\vec{x}_t - \vec{x}_{t-1})) + \alpha(\nabla f(\vec{x}_t, z_t) - \nabla F(\vec{x}_t))$$

so that the first term is now bounded by $O(\|\vec{x}_{t-1} - \vec{x}_t\|^2)$ which can be controlled through the learning rate $\eta$ and the second term is again easy to control via tuning $\alpha$. Without this second-order correction, we need to rely on $\hat{g}_{t-1} = \nabla F(\vec{x}_t) + O(\|\vec{x}_t - \vec{x}_{t-1}\|)$. Thus, by using another term of the Taylor expansion, we have improved the dependency to $O(\|\vec{x}_t - \vec{x}_{t-1}\|^2)$, which will create a corresponding reduction in our final error. Of course, the case $\hat{g}_{t-1} = \nabla F(\vec{x}_{t-1})$ above is examined for intuition only. In our analysis, we *do not* make any such assumption.

Note that this approach is morally similar to the methods of Cutkosky and Orabona [2019], Tran-Dinh et al. [2019], which are themselves similar to the recursive variance reduction Nguyen et al. [2017] algorithm for stochastic convex optimization. In these algorithms, the Hessian-vector product is replaced with two gradient evaluations with the same example $z_t$: $\nabla f(\vec{x}_t, z_t) - \nabla f(\vec{x}_{t-1}, z_t)$. So long as the individual functions $f(\vec{x}_t, z_t)$ are $L$-smooth, this will eventually have a similar correction effect. However, past empirical work suggests that variance reduction may not be effective in practice on deep learning tasks Defazio and Bottou [2018], while the use of Hessian-vector products has not been as extensively tested to our knowledge.

Our SGD with Hessian-corrected momentum is described in Algorithm 1, and its analysis is presented in Theorem 1. Note that we only need to make a small modification to the standard SGD update, which allows for streamlined analyses. Concretely, we avoid the large batch-size requirement of Arjevani et al. [2020a], and can extend the analysis to adaptive learning rates in Section 4.

---

**Algorithm 1** SGD with Hessian-corrected Momentum (**SGDHess**)

---

**Input:** Initial Point $\vec{x}_1$, learning rates $\eta_t$, momentum parameters $\alpha_t$, time horizon $T$, Lipschitz parameter $G$.
Sample $z_1 \sim P_z$.
$\hat{g}_1^{\text{clip}} \leftarrow \hat{g}_1 \leftarrow \nabla f(\vec{x}_1, z_1)$
$\vec{x}_2 \leftarrow \vec{x}_1 - \eta_1 \hat{g}_1^{\text{clip}}$
**for** $t = 2 \ldots T$ **do**
    Sample $z_t \sim P_z$.
    $\hat{g}_t \leftarrow (1 - \alpha_{t-1})(\hat{g}_{t-1}^{clip} + \nabla^2 f(\vec{x}_t, z_t)(\vec{x}_t - \vec{x}_{t-1})) + \alpha_{t-1} \nabla f(\vec{x}_t, z_t)$.
    $\hat{g}_t^{clip} \leftarrow \hat{g}_t$ if $\|\hat{g}_t\| \leq G$; otherwise, $\hat{g}_t^{clip} \leftarrow G \frac{\hat{g}_t}{\|\hat{g}_t\|}$
    $\vec{x}_{t+1} \leftarrow \vec{x}_t - \eta_t \hat{g}_t^{clip}$.
**end for**
Return $\hat{x}$ uniformly at random from $\vec{x}_1, \ldots, \vec{x}_T$ (in practice $\hat{x} = \vec{x}_T$).

---

**Theorem 1.** *Assume (1), (2), (4), (5), (6). Then Algorithm 1 with* $\eta_t = \frac{1}{Ct^{1/3}}$, $\alpha_t = 2K\eta_t\eta_{t+1}$ *with* $C \geq \sqrt{2K}$ *and* $C \geq 4L$, $K = \frac{2G^2\rho^2}{-2\sigma_H^2 + \sqrt{4\sigma_H^4 + \frac{\rho^2 G^2}{2}}}$ *and* $D = \frac{24}{5}K + \frac{16C^6}{25K^2}$ *guarantees*

$$\frac{1}{T}\sum_{t=1}^{T} \mathbb{E}[\|\nabla F(\vec{x}_t)\|^2] \leq \frac{20C\Delta + 96C^2G^2/K}{T^{2/3}} + \frac{20G^2D(1 + \log(T))}{C^2T^{2/3}}$$

To see why we want to use a clipped gradient, let us consider the error term with an *unclipped* gradient:

$$\hat{\epsilon}_{t+1} = (1 - \alpha_t)(\hat{g}_t - \nabla F(\vec{x}_t)) + (1 - \alpha_t)(\nabla^2 f(\vec{x}_{t+1}, z_{t+1}) - \nabla^2 F(\vec{x}_{t+1})(\vec{x}_{t+1} - \vec{x}_t))$$
$$+ (1 - \alpha_t)(\nabla F(\vec{x}_t) + \nabla^2 F(\vec{x}_{t+1})(\vec{x}_{t+1} - \vec{x}_t) - \nabla F(\vec{x}_{t+1})) + \alpha_t \epsilon_{t+1}^G$$

where $\epsilon_{t+1}^G = \nabla f(\vec{x}_{t+1}, z_{t+1}) - \nabla F(\vec{x}_{t+1})$. It is easy to see that we could bound the second and the fourth term using assumption (4) and (6) and the first term is simply $(1 - \alpha_t)\hat{\epsilon}_t$, which we can use to analyze how the error changes over each iteration . However, the third term is a bit trickier to control. Let $\delta_t = \nabla F(\vec{x}_t) + \nabla^2 F(\vec{x}_{t+1})(\vec{x}_{t+1} - \vec{x}_t) - \nabla F(\vec{x}_{t+1})$, from second-order smoothness we have:

$$\|\delta_t\|^2 \leq \frac{\rho^2}{4}\|\vec{x}_{t+1} - \vec{x}_t\|^4 \leq \frac{\rho^2}{4}\eta_t^4\|\hat{g}_t\|^4$$

Intuitively, if we let $\hat{g}_t$ be unbounded, $\|\hat{g}_t\|^4$ might be difficult to control since we only assume a bound on the variance of the $\nabla f(\vec{x}_t, z_t)$ rather than the fourth moment. Therefore, by enforcing some bound on the norm of $\hat{g}_t$ (using clipping), we make sure that we can control this term properly.

In order to prove this Theorem, we will require two Lemmas. The first (Lemma 2) is due to Cutkosky and Orabona [2019], and provides a bound on the progress of one iteration of stochastic gradient descent without making any assumptions (such as unbiasedness) about the gradient estimates. The second (Lemma 3), is a technical result characterizing the quality of the gradient estimates $\hat{g}_t^{clip}$ generated by Algorithm 1.

**Lemma 2.** *[Cutkosky and Orabona [2019] Lemma 1] Define:*

$$\hat{\epsilon}_t = \hat{g}_t^{clip} - \nabla F(\vec{x}_t)$$

*Suppose $\eta_t$ is a deterministic and non-increasing choice of learning rate. Then, so long as $\eta_t \leq \frac{1}{4L}$,*

$$\mathbb{E}[F(\vec{x}_{t+1}) - F(\vec{x}_t)] \leq -\frac{\eta_t}{4}\,\mathbb{E}[\|\nabla F(\vec{x}_t)\|^2] + \frac{3\eta_t}{4}\,\mathbb{E}[\|\hat{\epsilon}_t\|^2]$$

**Lemma 3.** *Suppose that $f(\vec{x}, z)$ satisfies (2), (4), (6), and (5). Define:*

$$\hat{\epsilon}_t = \hat{g}_t^{clip} - \nabla F(\vec{x}_t)$$

*Now, for some constant $C$ and $\sigma_H$, set $K = \dfrac{2G^2\rho^2}{-2\sigma_H^2 + \sqrt{4\sigma_H^4 + \frac{\rho^2 G^2}{2}}}$, $\eta_t = \frac{1}{Ct^{1/3}}$ with $C \geq \sqrt{2K}$, and $\alpha_t = 2K\eta_t\eta_{t+1}$. Then we have:*

$$\frac{6}{5K\eta_{t+1}}\,\mathbb{E}[\|\hat{\epsilon}_{t+1}\|^2] - \frac{6}{5K\eta_t}\,\mathbb{E}[\|\hat{\epsilon}_t\|^2] \leq -\frac{3\eta_t}{4}\,\mathbb{E}[\|\hat{\epsilon}_t\|^2] + \frac{\eta_t}{5}\,\mathbb{E}[\|\nabla F(\vec{x}_t)\|^2] + \eta_t^3\left(\frac{24}{5}KG^2 + \frac{16C^6 G^2}{25K^2}\right)$$

Let us look into how Lemma 3 is used in our analysis. To prove Theorem 1, we use the Lyapunov function defined as $\Phi_t = F(\vec{x}_t) + \frac{6}{5K\eta_t}\|\hat{\epsilon}_t\|^2$ to bound the $\mathbb{E}[\|\nabla F(\vec{x}_t)\|^2]$ term. Then:

$$\mathbb{E}[\Phi_{t+1} - \Phi_t] = \mathbb{E}\left[F(\vec{x}_{t+1}) - F(\vec{x}_t) + \frac{6}{5K\eta_{t+1}}\|\hat{\epsilon}_{t+1}\|^2 - \frac{6}{5K\eta_t}\|\hat{\epsilon}_t\|^2\right]$$

$$\leq \mathbb{E}\left[-\frac{\eta_t}{4}\|\nabla F(\vec{x}_t)\|^2 + \frac{3\eta_t}{4}\|\hat{\epsilon}_t\|^2 + \frac{6}{5K\eta_{t+1}}\|\hat{\epsilon}_{t+1}\|^2 - \frac{6}{5K\eta_t}\|\hat{\epsilon}_t\|^2\right] \quad (7)$$

where the inequality comes from Lemma 2. Now if we plug in the result of Lemma 3, we are able to simplify the bound of (7) by canceling the positive error term $\frac{3\eta_t}{4}\|\hat{\epsilon}_t\|^2$ while keeping the coefficient of $\|\nabla F(\vec{x}_t)\|^2$ negative. Then, we can move the negative term to the left hand side and derive a bound for $\mathbb{E}[\|\nabla F(\vec{x}_t)\|^2]$. With Lemma 2 and Lemma 3 in hand, we are now ready to prove Theorem 1.

*Proof of Theorem 1.* We define the potential:

$$\Phi_t = F(\vec{x}_t) + \frac{6}{5K\eta_t}\|\hat{\epsilon}_t\|^2$$

We will then show that $\Phi_t$ roughly decreases with $t$ at rate that depends on $\|\nabla F(\vec{x}_t)\|^2$. Specifically:

$$\mathbb{E}[\Phi_{t+1} - \Phi_t] = \mathbb{E}\left[F(\vec{x}_{t+1}) - F(\vec{x}_t) + \frac{6}{5K\eta_{t+1}}\|\hat{\epsilon}_{t+1}\|^2 - \frac{6}{5K\eta_t}\|\hat{\epsilon}_t\|^2\right]$$

applying Lemmas 2 and 3:

$$\leq -\frac{\eta_t}{4}\,\mathbb{E}[\|\nabla F(\vec{x}_t)\|^2] + \frac{3\eta_t}{4}\,\mathbb{E}[\|\hat{\epsilon}_t\|^2] - \frac{3\eta_t}{4}\,\mathbb{E}[\|\hat{\epsilon}_t\|^2] + \frac{\eta_t}{5}\,\mathbb{E}[\|\nabla F(\vec{x}_t)\|^2]$$

$$+ \eta_t^3 G^2\left(\frac{24}{5}K + \frac{16C^6}{25K^2}\right)$$

$$= \frac{-\eta_t}{20}\,\mathbb{E}[\|\nabla F(\vec{x}_t)\|^2] + \eta_t^3 G^2\left(\frac{24}{5}K + \frac{16C^6}{25K^2}\right)$$

Let $D = \frac{24}{5}K + \frac{16C^6}{25K^2}$:

$$= \frac{-\eta_t}{20}\,\mathbb{E}[\|\nabla F(\vec{x}_t)\|^2] + \eta_t^3 G^2 D$$

Now sum over $t$, and use $\eta_t \geq \eta_T$ for all $t$:

$$\mathbb{E}[\Phi_{T+1} - \Phi_1] \leq \frac{-\eta_T}{20}\sum_{t=1}^{T}\mathbb{E}[\|\nabla F(\vec{x}_t)\|^2] + G^2 D\sum_{t=1}^{T}\eta_t^3$$

$$\Leftrightarrow \sum_{t=1}^{T}\mathbb{E}[\|\nabla F(\vec{x}_t)\|^2] \leq \frac{20}{\eta_T}\mathbb{E}[\Phi_1 - \Phi_{T+1}] + \frac{20G^2 D\sum_{t=1}^{T}\eta_t^3}{\eta_T}$$

Now, observe that:

$$\mathbb{E}[\Phi_1 - \Phi_{T+1}] = \mathbb{E}\left[F(\vec{x}_1) - F(\vec{x}_{T+1}) + \frac{6}{5K\eta_1}\|\hat{\epsilon}_1\|^2 - \frac{6}{5K\eta_{T+1}}\|\hat{\epsilon}_{T+1}\|^2\right]$$

$$\leq \Delta + \frac{24G^2}{5K\eta_1} \leq \Delta + \frac{24CG^2}{5K}$$

where in the second line we used $\mathbb{E}[\|\hat{\epsilon}_1\|^2] \leq 4G^2$. Also, we have:

$$\sum_{t=1}^{T}\eta_t^3 = \frac{1}{C^3}\sum_{t=1}^{T}\frac{1}{t} \leq \frac{1 + \log(T)}{C^3}$$

Putting all this together yields:

$$\frac{1}{T}\sum_{t=1}^{T}\mathbb{E}[\|\nabla F(\vec{x}_t)\|^2] \leq \frac{20}{T\eta_T}\mathbb{E}[\Phi_1 - \Phi_{T+1}] + \frac{20G^2 D\sum_{t=1}^{T}\eta_t^3}{T\eta_T}$$

$$\leq \frac{20C\Delta + 96C^2 G^2/K}{T^{2/3}} + \frac{20G^2 D(1 + \log(T))}{C^2 T^{2/3}}$$

$\square$

## 3 Normalized SGD with Hessian-corrected momentum

---
**Algorithm 2** Normalized SGD with Hessian-corrected Momentum (**N-SGDHess**)

---
**Input:** Initial Point $\vec{x}_1$, learning rates $\eta$, momentum parameters $\alpha$, time horizon $T$:
Sample $z_1 \sim P_z$.
$\hat{g}_1 \leftarrow \nabla f(\vec{x}_1, z_1)$.
$\vec{x}_2 \leftarrow \vec{x}_1 - \eta\frac{\hat{g}_1}{\|\hat{g}_1\|}$
**for** $t = 2 \ldots T$ **do**
   Sample $z_t \sim P_z$.
   $\hat{g}_t \leftarrow (1-\alpha)(\hat{g}_{t-1} + \nabla^2 f(\vec{x}_t, z_t)(\vec{x}_t - \vec{x}_{t-1})) + \alpha\nabla f(\vec{x}_t, z_t)$.
   $\vec{x}_{t+1} \leftarrow \vec{x}_t - \eta\frac{\hat{g}_t}{\|\hat{g}_t\|}$.
**end for**
Return $\hat{x}$ uniformly at random from $\vec{x}_1, \ldots, \vec{x}_T$ (in practice $\hat{x} = \vec{x}_T$).

---

In this section, we introduce an algorithm that dispenses with the assumption (6) required by Algorithm 1. This method (Algorithm 2) uses SGD with normalized updates and Hessian-vector product-based momentum. We will show that normalization can significantly simplify the analysis from section 2 while still maintaining $O(\epsilon^{-3})$ convergence rate. Indeed, the technical term in the analysis of Algorithm 1 that required us to enforce a bound on the updates via clipping simply does not appear because the updates are automatically bounded by $\eta$.

The new convergence bound is presented in Theorem 4.

**Theorem 4.** *Assuming (1), (2), (3), (4), and (5) hold (but* not *assuming (6)), with* $\alpha = \min\{\max\{\frac{1}{T^{2/3}}, \frac{\Delta^{4/5}\rho^{2/5}}{T^{4/5}\sigma_G^{6/5}}, \frac{(2\Delta\sigma_H)^{2/3}}{T^{2/3}\sigma_G^{4/3}}\}, 1\}$ *and* $\eta = \min\{\frac{\sqrt{2\Delta}\alpha^{1/4}}{\sqrt{T(L\sqrt{\alpha}+4\sigma_H)}}, \frac{(\Delta\alpha)^{1/3}}{(\rho T)^{1/3}}\}$, *Algorithm 2 guarantees*

$$\mathbb{E}[\|\nabla F(\vec{x}_t)\|] \leq \frac{6\sigma_G + 54^{1/3}(\Delta\sigma_H)^{1/3}\sigma_G^{1/3}}{T^{1/3}} + \frac{6\sigma_G^{2/5}\Delta^{2/5}\rho^{1/5}}{T^{2/5}} + \frac{\sqrt{9\Delta L} + \sqrt{72\Delta\sigma_H}}{\sqrt{T}} + \frac{6\Delta^{2/3}\rho^{1/3}}{T^{2/3}}$$

$$+ \frac{\sqrt{18\Delta\sigma_H}}{T^{3/2}} + \frac{3\Delta^{2/3}\rho^{1/3}}{T^{5/3}}$$

*In words, Algorithm 2 achieves* $O(1/T^{1/3})$ *with large* $\sigma_H$ *and* $\sigma_G$, *and achieves* $O(1/\sqrt{T})$ *in noiseless case, without requiring a Lipschitz bound on the objective.*

The proof of Theorem 4 is provided in the appendix.

## 4 Adaptive SGD with Hessian-corrected Momentum

---
**Algorithm 3** Adaptive learning rate for SGD with Hessian-corrected Momentum

---
**Input:** Initial Point $\vec{x}_1$, parameters $c$, $w$, $\alpha_t$, time horizon $T$, parameter $G$:
Sample $z_1 \sim P_z$.
$\hat{g}_1^{clip} \leftarrow \hat{g}_1 \leftarrow \nabla f(\vec{x}_1, z_1)$
$G_1 \leftarrow \|\nabla f(\vec{x}_1, z_1)\|$.
$\eta_1 \leftarrow \frac{c}{w^{1/3}}$
$\vec{x}_2 \leftarrow \vec{x}_1 - \eta_1 \hat{g}_1^{clip}$
**for** $t = 2 \ldots T$ **do**
    Sample $z_t \sim P_z$.
    $G_1 \leftarrow \|\nabla f(\vec{x}_t, z_t)\|$
    $\hat{g}_t \leftarrow (1 - \alpha_{t-1})(\hat{g}_{t-1}^{clip} + \nabla^2 f(\vec{x}_t, z_t)(\vec{x}_t - \vec{x}_{t-1})) + \alpha_{t-1}\nabla f(\vec{x}_t, z_t)$.
    $\hat{g}_t^{clip} \leftarrow \hat{g}_t$ if $\|\hat{g}_t\| \leq G$; otherwise, $\hat{g}_t^{clip} \leftarrow G\frac{\hat{g}_t}{\|\hat{g}_t\|}$
    $\eta_t \leftarrow \frac{c}{(w+\sum_{i=1}^{t-2} G_i^2)^{1/3}}$    (set $\eta_2 = \frac{c}{w^{1/3}}$).
    $\vec{x}_{t+1} \leftarrow \vec{x}_t - \eta_t \hat{g}_t^{clip}$.
**end for**
Return $\hat{x}$ uniformly at random from $\vec{x}_1, \ldots, \vec{x}_T$ (in practice $\hat{x} = \vec{x}_T$).

---

In previous sections, we have presented two different versions of our algorithm. Both algorithms achieve the worst-case optimal $O(1/T^{1/3})$ convergence rate. However, the bound in Section 2 is non-adaptive and remains to be $O(1/T^{1/3})$ even in noiseless settings (e.g. if $\sigma_G = 0$). The bound for the normalized SGD algorithm of Section 3 is better in the sense that when the noise is negligible, the algorithm can achieve $O(1/\sqrt{T})$ with explicit tuning of the learning rate, but this requires us to set the parameters $\eta$ and $\alpha$ based on prior knowledge of $\sigma_G$. In this section, we will describe an *adaptive* version of our algorithm that has the best of both worlds (Algorithm 3). The algorithm does not require the knowledge of $\sigma_G$ but still automatically improves to a tighter bound whenever $\sigma_G$ is small.

**Theorem 5.** *With* $K = \frac{2G^2\rho^2}{-4\sigma_H^2+\sqrt{16\sigma_H^4+\frac{\rho^2 G^2}{2}}}$, $\eta_t = \frac{c}{(w+\sum_{i=1}^{t-2} G_i^2)^{1/3}}$ *with* $c \leq \frac{2G^{2/3}}{\sqrt{K}}$ *and* $w = \max\{(4Lc)^3, 3G^2\}$, $\alpha_t = 2K\eta_t\eta_{t+1}$. *Algorithm 3 guarantees:*

$$\mathbb{E}\left[\frac{1}{T}\sum_{t=1}^{T}\|\nabla F(\vec{x}_t)\|\right] \leq \frac{w^{1/6}\sqrt{2M} + 2M^{3/4}}{\sqrt{T}} + \frac{2\sigma_G^{1/3}}{T^{1/3}}$$

*Where* $M = \frac{1}{c}(20(\Delta + \frac{6\sigma_G^2 w^{1/3}}{5Kc}) + 192Kc^2\ln(T+1) + \frac{64G^4}{5K^2c^3}\ln T)$.

In words, algorithm 3 converges with $O\left(\frac{1}{T^{1/3}}\right)$ rate in noisy case and automatically improves to $\tilde{O}\left(\frac{1}{\sqrt{T}}\right)$ in noiseless case. We defer the proof of Theorem 5 to the appendix.

# 5 Experiments

## 5.1 Setup

Our algorithm is a simple extension of the official SGD implementation in Pytorch. The Hessian-vector products can be efficiently computed using the automatic differentiation package Paszke et al. [2017]: $\nabla^2 f(\vec{x}, z)v = \nabla h(\vec{x})$ where $h(\vec{x}) = \langle \nabla f(\vec{x}, z), v \rangle$. Since Pytorch allows backprogation through the differentiation process itself, this is straightforward to implement. To validate the effectiveness of our proposed algorithm, we perform experiments in two tasks: image classification and neural machine translation on popular deep learning benchmarks. The performance of SGDHess (unclipped version - we found that the clipped version had similar results but was slightly slower) is compared to those of commonly used optimizers such as Adam and SGD as well as AdaHessian, another algorithm that incorporates second-order information. All experiments are run on NVIDIA v100 GPUs. The link to the code is provided in the appendix.

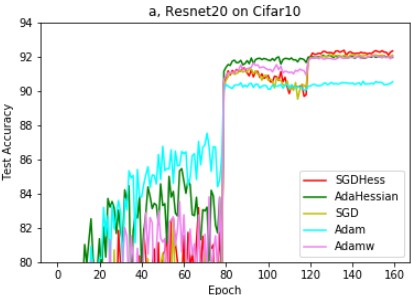 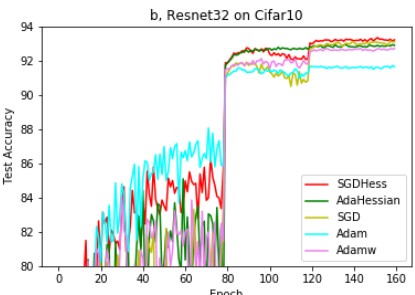

Figure 1: (a) SGDHess (Red) achieves the best accuracy among all optimizers (92.46 $\pm 0.07\%$). (b) Similar to the experiment on Resnet20, SGDHess also converges to the best accuracy (93.19 $\pm 0.08\%$). For the full plot, refer to the appendix.

## 5.2 Image Classification

Our Cifar10 experiment is conducted using the official implementation of AdaHessian. For AdaHessian, we use their recommended values for the same task for all the parameters. For the rest of the optimizers, we performed a grid search on the base learning rate $\eta \in \{0.001, 0.005, 0.01, 0.05, 0.1, 0.5, 1\}$ to find the best settings. Similar to the Cifar10 experiment of AdaHessian, we also trained our models on 160 epochs and we ran each optimizer 5 times and reported the average best accuracy as well as the standard deviation (detailed results in the appendix). As we can see from the results in Figure 1, SGDHess outperforms all other optimizers (0.32% and 0.11% better than the next best optimizer in Resnet20 and Resnet32 respectively).

Table 1: Top 1 accuracy on Imagenet

| SGD | SGDHess | AdaHessian |
|---|---|---|
| 70.36 | 70.58 | 69.89 |

We also train SGD, SGDHess, and AdaHessian with Imagenet Deng et al. [2009] on Resnet18 to see how well SGDHess perform on a larger-scale benchmark. We use standard parameter values for SGD (lr = 0.1, momentum = 0.9, weight_decay = 1e-4) for both SGD and SGDHess and the recommended parameters values for AdaHessian. For the learning rate scheduler, we employ the plateau decay scheduler that was used in Yao et al. [2020]. We train our model in 90 epochs as usual. Even without extensive tuning, SGDHess not only still outperforms SGD (as shown in Table 1) but also comes close to the state-of-the-art accuracy (70.7%) on this particular task Redmon [2013–2016], even though these settings were chosen with SGD in mind rather than SGDHess.

Table 2: Bleu Score on IWSLT'14

| SGD | SGDHess | AdamW | AdaHessian |
|---|---|---|---|
| 29.75 | 33.73 | 33.95 | 33.62 |

## 5.3 Neural Machine Translation

We use the IWSLT'14 German to English dataset that contains 153k/7k/7k in the train/validation/test set. Our experiments are run using all the default values specified in the official implementation of Fairseq Ott et al. [2019]. We use BLEU Papineni et al. [2002] as the evaluation metrics for our experiment. For AdaHessian, we use the parameters specified in Yao et al. [2020]. For other optimizers, we again run a grid search to find the best learning rate for the task. The best bleu scores are reported in Table 2. It is worth stressing that SGDHess is an algorithm based on SGD, which consistently performs much worse than adaptive algorithms such as AdaHessian and AdamW in this type of task. Still, SGDHess is able to produce comparable results to those of AdaHessian and AdamW, thus significant bridging the gap between SGD and other adaptive algorithms (an almost 4 points increase compared to SGD's in BLEU score, which is significant for the task).

## 6 Conclusion and Future Work

In this paper, we have presented SGDHess, a novel SGD-based algorithm using Hessian-corrected momentum. We show that when the objective is second-order smooth, our algorithm can achieve the optimal $O(\epsilon^{-3})$ bound, and we provide a variant that automatically adapts to the level of noise in the gradients. Finally, we provide experimental results on computer vision and neural machine translation. In both tasks, SGDHess consistently performs better or comparable to other commonly used optimizers such as SGD and Adam. It is our hope that this work demonstrates that Hessian-based optimization can combine both theoretical and practical improvements for large-scale machine learning problems. Our algorithm is one of the first second-order method that is able to achieve the optimal theoretical convergence rate and compete with first-order methods on large-scale tasks in terms of performance and runtime.

**Limitations:** Our modification to the standard momentum formula is also extremely simple, and so we suspect that is possible to make similar modifications to other popular optimization algorithms that use momentum, but we did not investigate this possibility. For example, one might hope that an appropriate per-coordinate or per-layer adaptaptation a la Adam Kingma and Ba [2014], AMSGrad Reddi et al. or LAMB You et al. [2019] might improve performance. On the theoretical side, our adaptive results require Lipschitz losses, which may not be truly necessary.

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
