# A Appendix

## A.1 License

Image Classification: Imagenet has BSD 3-Clause License, Resnet has Apache License, Cifar10 has MIT License.

Neural Machine Translation: Fairseq has MIT License.

All experiments are implemented on Pytorch which has BSD License. Other assets that we use have no license.

## A.2 Additional details on the Experiments

All the implementations are publicly available here: https://github.com/tranhp98/SGDHess.

**Image Classification:** Here we provide some extra details of our experiments. From the results in Table 3, we can see that SGDHess achieves the best accuracy among all optimizers. SGDHess also has the lowest standard deviation, indicating that it consistently performs well in all experiments. For Imagenet task, our code is based on the official implementation of Imagenet on Pytorch. We also keep all the default settings constant. The only thing that we change is the learning rate schedule (from step decay every 30 epochs to plateau decay where we decrease our learning rate by a factor of two if we do not make progress in three consecutive epochs) based on the suggestion from Yao et al. [2020]. All the experiments are run with batch size = 256.

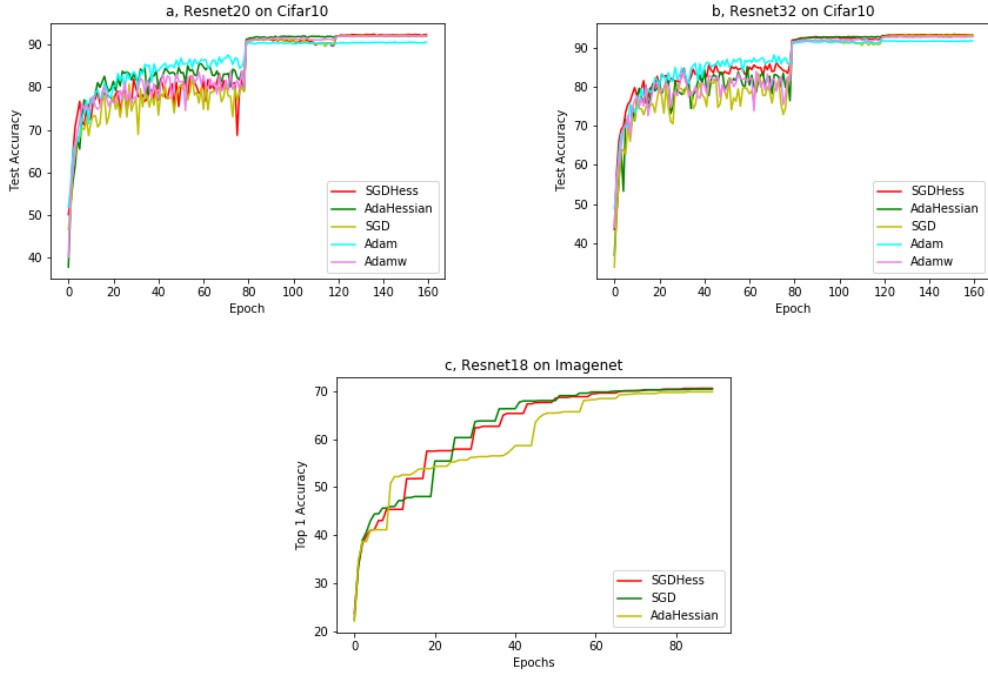

Figure 2: (a) Test accuracy on Resnet20 (b) Test accuracy on Resnet32 (c) Test accuracy on Imagenet

**Comparison to STORM:** We also run experiments with STORM to compare its performance to SGDHess. For STORM, we use a grid search to tune the three hyperparameters k, c, w and remove the learning rate schedule that we use in the previous experiments with other optimizers. The reason for this is STORM already has its own learning rate decay mechanism and its performance actually got worse when we tested it with learning rate schedule (this could be resulted from the interplay between learning rate and momentum of STORM). As we can see from the figure, the performance of STORM is not very competitive with SGDHess. The best accuracy the STORM achieves in

Table 3: Test accuracy on Cifar10

|  | Resnet20 | Resnet32 |
|---|---|---|
| SGD | $92.14 \pm 0.15$ | $93.08 \pm 0.12$ |
| AdaHessian | $92.11 \pm 0.07$ | $92.96 \pm 0.09$ |
| Adam | $90.58 \pm 0.28$ | $91.62 \pm 0.12$ |
| AdamW | $92.05 \pm 0.15$ | $92.45 \pm 0.25$ |
| SGDHess | $92.46 \pm 0.07$ | $93.19 \pm 0.08$ |

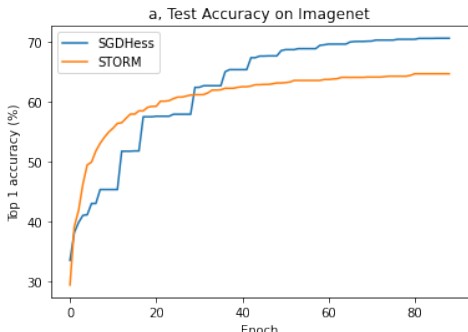 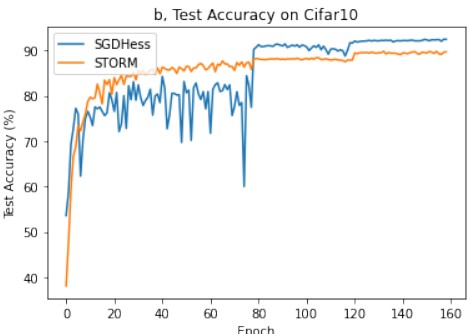

Figure 3: (a) Comparison in Imagenet (b) Comparison in Cifar10

Imagenet and Cifar10 tasks are 64.65% and 89.83% respectively (compared to 70.56% and 92.47% of SGDHess).

**Neural Machine Translation:** The settings of our experiments follow exactly the settings specified in the translation examples in Ott et al. [2019]. The only things that we tune are learning rate, weight decay, and number of updates if needs be. Other than the main results reported in Table 2, we also run extra experiment with our adaptive algorithm described in Section 4. The adaptive algorithm achieves the best BLEU score of 35.53, which is slightly worse than the non-adaptive algorithm. The main advantage of the adaptive algorithm is that it is a bit less sensitive to the change in the learning rate. For the non-adaptive algorithm, we need to warm up our optimizer very gradually (we set the number of updates to 8000) else we would run into exploding gradients problem. On the other hand, for the adaptive one, we can just set the default number of updates without any problems. We also suspect that incorporating the per-coordinate or diagonal-style adaptivity of popular optimizers such as Adam may provide an useful future direction for improvement.

**Discussion on run time and memory.** Since SGDHess requires the computation of Hessian-vector product in every iteration to "correct" the momentum, it is inevitable that its run time is slower than that of first-order algorithms such as Adam or SGD. Fortunately, the penalty is small even in our unoptimized implementation. Specifically, for image classification task, SGDHess is roughly 1.7/1.6 times slower than SGD and Adam respectively (AdaHessian is 1.9/1.7 times slower than SGD/Adam). For the NLP task, SGDHess is 1.3 times slower than SGD/Adam (AdaHessian is 1.7 times slower). Furthermore, the tuning of SGDHess is relatively straighforward: in many cases, the optimal tuning of SGDHess is the same as that of SGD. Thus, to reduce the computation overhead, one could try tuning SGD first then using the optimal parameters of SGD for SGDHess. In term of memory overhead, SGDHess used roughly 10% extra memory compared to SGD in Imagenet task. However, this issue can be easily bypassed using parallel programming. Since our implementation was not optimized for speed or memory, we suspect that there are improvements that can be made which we leave for future work.

**Comparison to Variance-Reduction.** Our algorithm bears a resemblance to some variance-reduction based algorithms, most notably STORM Cutkosky and Orabona [2019], in the sense that both approaches add some extra term to "correct" the bias in the momentum. However, while STORM makes

the typical variance-reduction assumption that each $f(\vec{x}_t, z_t)$ is smooth (per-example smoothness)[2], SGDHess only requires $F(x) = \mathbb{E}_{z_t}[f(\vec{x}_t, z_t)]$ to be smooth (on-average smoothness), which is a weaker assumption and can be applied to a wider range of functions. Furthermore, the convergence rate of STORM depends on this per-example smoothness constant, which is larger than the on-average smoothness constant in the bound of SGDHess. To illustrate this difference, we performed

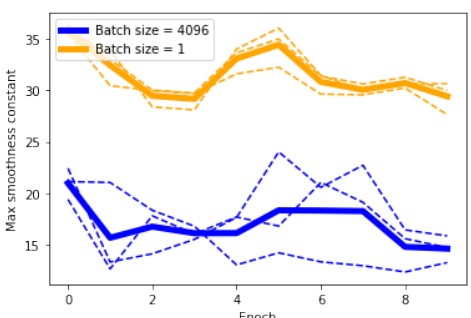

Figure 4: Largest smoothness constant per epoch (Cifar10/Resnet 32). Dotted lines represent individual training runs, thick is the average.

some experiments to approximate these smoothness constants. We train Resnet32 on Cifar10 using SGD with learning rate = 0.1 for 10 epochs with batch size = 1 (to approximate the per-example smoothness needed in variance reduction) and batch size = 4096 (to approximate the on-average smoothness needed by SGDHess). To obtain the true on-average smoothness constant we would need a full-batch rather than a minibatch, so our estimate here is *larger* than the true value. We compute an approximate smoothness constant at each iteration by computing the ratio $\frac{\|\nabla f(\vec{x}_t, z_t) - \nabla f(\vec{x}_{t+1}, z_t)\|}{\|x_t - x_{t+1}\|}$ (or its minibatch equivalent with both gradients replaced with averages) and record the largest such ratio encountered in every epoch in Figure 4. In the figure, the orange and blue dotted lines indicate the smoothness constant of each run with batch size = 1 and 4096 respectively. The thicker lines are the average of all 3 runs. From the graph, we can clearly see that the maximum smoothness constant is consistently larger (by about 2X) when we evaluate on a single example rather then on a large sample.

Another advantage of SGDHess over STORM and other SVRG-like algorithms is the ease of implementation and deployment. SVRG-based algorithms require two gradient evaluations at different weight values but the *same* minibatch. This means that typically in order to implement such an algorithm one must build a custom training loop rather than using any built-in methods. The situation is even more intricate when the model incorporates inherent randomness like drop-out, as this requires both gradient evaluations to keep the same dropout mask. In contrast, implementing a hessian-vector product is quite easy and can be done purely inside the optimizer without modifying external code.

Finally, it is unclear how effective variance reduction based algorithms are for deep learning in practice Defazio and Bottou [2018]. In particular, STORM has not been tested in a wide variety of deep learning tasks and in our own experiments it was not competitive (we include these results in the appendix). STORM also has slightly higher run time than SGDHess. For Imagenet, the run time per epoch of SGDHess and STORM were 36 minutes and 44 minutes respectively.

## A.3 Supplemental Lemmas

**Lemma 6.** *Let* $X = \min\{\frac{\sqrt{A}}{\sqrt{B}}, \frac{A^{1/3}}{C^{1/3}}\}$. *Then:*

$$\frac{A}{X} + BX + CX^2 \leq 2\sqrt{AB} + 2A^{2/3}C^{1/3}$$

---

[2]Technically, one need only require $\mathbb{E}[\|\nabla f(\vec{x}_t, z_t) - \nabla f(\vec{x}_{t+1}, z_t)\|^2] \leq \mathbb{E}[L\|\vec{x}_t - \vec{x}_{t+1}\|^2]$ if we are willing to sacrifice adaptive convergence rates.

*Proof.* Bound the first term:

$$\frac{A}{X} \le \max\{\sqrt{AB}, A^{2/3}C^{1/3}\} \le \sqrt{AB} + A^{2/3}C^{1/3}$$

Let us do some case work for the second term. If $X = \frac{\sqrt{A}}{\sqrt{B}}$ then $BX = \sqrt{AB}$. Otherwise, if $X = \frac{A^{1/3}}{C^{1/3}}$, then $\frac{A^{1/3}}{C^{1/3}} \le \frac{\sqrt{A}}{\sqrt{B}} \Rightarrow \frac{\sqrt{B}}{C^{1/3}} \le A^{1/6}$. Therefore,

$$BX = \frac{BA^{1/3}}{C^{1/3}} \le \sqrt{B}A^{1/6}A^{1/3}$$
$$\le \sqrt{AB}$$

In either case, $BX \le \sqrt{AB}$. Now repeat the same arguments for $CX^2$ term, we would get $CX^2 \le A^{2/3}C^{1/3}$. Then we can combine the bounds to get the desired result. $\square$

**Lemma 7.** *Let* $X = \max\{\frac{B^{2/3}}{A^{2/3}}, \frac{C^{6/5}}{A^{6/5}}, \frac{D^{4/3}}{A^{4/3}}\}$. *Then:*

$$A\sqrt{X} + \frac{B}{X} + \frac{C}{X^{1/3}} + \frac{D}{X^{1/4}} \le 2(B^{1/3}A^{2/3} + C^{3/5}A^{2/5}$$
$$+ D^{2/3}A^{1/3})$$

*Proof.* The proof is almost the same as the proof of Lemma 6. First, we bound the first term:

$$A\sqrt{X} \le \max\{\frac{B^{1/3}}{A^{2/3}}, \frac{C^{3/5}}{A^{2/5}}, \frac{D^{2/3}}{A^{1/3}}\} \le B^{1/3}A^{2/3} + C^{3/5}A^{2/5}$$
$$+ D^{2/3}A^{1/3}$$

Then we can do some case works for the other 3 terms, which would get us $\frac{B}{X} + \frac{C}{X^{1/3}} + \frac{D}{X^{1/4}} \le B^{1/3}A^{2/3} + C^{3/5}A^{2/5} + D^{2/3}A^{1/3}$. Now combining the bounds to get the desired results. $\square$

**Lemma 8.** *With* $\alpha = \min\{\max\{\frac{1}{T^{2/3}}, \frac{\Delta^{4/5}\rho^{2/5}}{T^{4/5}\sigma_G^{6/5}}, \frac{(2\Delta\sigma_H)^{2/3}}{T^{2/3}\sigma_G^{4/3}}\}, 1\}$ *and* $\eta = \min\{\frac{\sqrt{2\Delta}\alpha^{1/4}}{\sqrt{T(L\sqrt{\alpha}+4\sigma_H)}}, \frac{(\Delta\alpha)^{1/3}}{(\rho T)^{1/3}}\}$, *we have:*

$$\frac{3\Delta}{2\eta T} + \frac{3L\eta}{4} + \frac{3\sigma_G}{\alpha T} + \frac{3\eta^2\rho}{2\alpha} + \frac{3\eta\sigma_H}{\sqrt{\alpha}} + 3\sigma_G\sqrt{\alpha}$$
$$\le \frac{6\sigma_G + 54^{1/3}(\Delta\sigma_H)^{1/3}\sigma_G^{1/3}}{T^{1/3}} + \frac{6\sigma_G^{2/5}\Delta^{2/5}\rho^{1/5}}{T^{2/5}}$$
$$+ \frac{\sqrt{9\Delta L} + \sqrt{72\Delta\sigma_H}}{\sqrt{T}} + \frac{6\Delta^{2/3}\rho^{1/3}}{T^{2/3}} + \frac{\sqrt{18\Delta\sigma_H}}{T^{3/2}} + \frac{3\Delta^{2/3}\rho^{1/3}}{T^{5/3}}$$

*Proof.* Let $F = \frac{3\Delta}{2\eta T} + \frac{3L\eta}{4} + \frac{3\sigma_G}{\alpha T} + \frac{3\eta^2\rho}{2\alpha} + \frac{3\eta\sigma_H}{\sqrt{\alpha}} + 3\sigma_G\sqrt{\alpha}$. Applying Lemma 6 with $A = \frac{3\Delta}{2T}$, $B = \frac{3L}{4} + \frac{3\sigma_H}{\sqrt{\alpha}}$, and $C = \frac{3\rho}{2\alpha}$ and $\eta$ set to the value of $X$ specified by the Lemma:

$$F \le 3\sqrt{2}\sqrt{\frac{\Delta}{T}(\frac{L}{4} + \frac{\sigma_H}{\sqrt{\alpha}})} + \frac{3\Delta^{2/3}\rho^{1/3}}{T^{2/3}\alpha^{1/3}} + \frac{3\sigma_G}{\alpha T} + 3\sigma_G\sqrt{\alpha}$$

Use $\sqrt{a+b} \le \sqrt{a} + \sqrt{b}$:

$$F \le \frac{3}{\sqrt{2}}\frac{\sqrt{\Delta L}}{\sqrt{T}} + 3\sqrt{2}\frac{\sqrt{\Delta\sigma_H}}{\sqrt{T}\alpha^{1/4}} + \frac{3\Delta^{2/3}\rho^{1/3}}{T^{2/3}\alpha^{1/3}} + \frac{3\sigma_G}{\alpha T}$$
$$+ 3\sigma_G\sqrt{\alpha} \quad (8)$$

Now applying Lemma 7 with $X = \alpha$, $A = 3\sigma_G$, $B = \frac{3\sigma_G}{T}$, $C = \frac{3\Delta^{2/3}\rho^{1/3}}{T^{2/3}}$, and $D = 3\sqrt{2}\frac{\sqrt{\Delta\sigma_H}}{\sqrt{T}}$:

$$F \leq \frac{3\sqrt{\Delta L}}{\sqrt{2}\sqrt{T}} + \frac{6\sigma_G}{T^{1/3}} + \frac{6\sigma_G^{2/5}\Delta^{2/5}\rho^{1/5}}{T^{2/5}}$$
$$+ \frac{54^{1/3}(\Delta\sigma_H)^{1/3}\sigma_G^{1/3}}{T^{1/3}}$$

Let $\alpha = \min\{\max\{\frac{1}{T^{2/3}}, \frac{\Delta^{4/5}\rho^{2/5}}{T^{4/5}\sigma_G^{6/5}}, \frac{(2\Delta\sigma_H)^{2/3}}{T^{2/3}\sigma_G^{4/3}}\}, 1\}$. Since $\frac{1}{T^{2/3}} \leq 1$, let us examine the other two cases when $\alpha \geq 1$.

*Case 1:* $\frac{\Delta^{4/5}\rho^{2/5}}{T^{4/5}G^{6/5}} \geq 1$. Then we have $\sigma_G \leq \frac{\Delta^{2/3}\rho^{1/3}}{T^{2/3}}$. Substitute to (8) with $\alpha = 1$:

$$F \leq \frac{3\sqrt{\Delta L}}{\sqrt{2}\sqrt{T}} + \frac{3\sqrt{2}\sqrt{\Delta\sigma_H}}{\sqrt{T}} + \frac{3\Delta^{2/3}\rho^{1/3}}{T^{2/3}} + \frac{3\sigma_G}{T} + 3\sigma_G$$
$$\leq \frac{3\sqrt{\Delta L}}{\sqrt{2}\sqrt{T}} + \frac{3\sqrt{2}\sqrt{\Delta\sigma_H}}{\sqrt{T}} + \frac{3\Delta^{2/3}\rho^{1/3}}{T^{2/3}} + \frac{3\Delta^{2/3}\rho^{1/3}}{T^{5/3}} + \frac{3\Delta^{2/3}\rho^{1/3}}{T^{2/3}}$$

*Case 2:* $\frac{(2\Delta\sigma_H)^{2/3}}{T^{2/3}\sigma_G^{4/3}} \geq 1$. Then we have $\sigma_G \leq \frac{\sqrt{2\Delta\sigma_H}}{\sqrt{T}}$. Substitute to (8) with $\alpha = 1$:

$$F \leq \frac{3\sqrt{\Delta L}}{\sqrt{2}\sqrt{T}} + \frac{3\sqrt{2}\sqrt{\Delta\sigma_H}}{\sqrt{T}} + \frac{3\Delta^{2/3}\rho^{1/3}}{T^{2/3}} + \frac{3\sqrt{2\Delta\sigma_H}}{T^{3/2}} + \frac{3\sqrt{2\Delta\sigma_H}}{\sqrt{T}}$$

Now combine the bounds to get the desired result. $\qquad\square$

## A.4 Proof of section 2

**Lemma 2.** *[Cutkosky and Orabona [2019] Lemma 1] Define:*

$$\hat{\epsilon}_t = \hat{g}_t^{clip} - \nabla F(\vec{x}_t)$$

*Suppose $\eta_t$ is a deterministic and non-increasing choice of learning rate. Then, so long as $\eta_t \leq \frac{1}{4L}$,*

$$\mathbb{E}[F(\vec{x}_{t+1}) - F(\vec{x}_t)] \leq -\frac{\eta_t}{4}\mathbb{E}[\|\nabla F(\vec{x}_t)\|^2] + \frac{3\eta_t}{4}\mathbb{E}[\|\hat{\epsilon}_t\|^2]$$

*Proof.*

$$F(\vec{x}_{t+1}) \leq F(\vec{x}_t) + \langle \nabla F(\vec{x}_t), \vec{x}_{t+1} - \vec{x}_t \rangle + \frac{L}{2}\|\vec{x}_{t+1} - \vec{x}_t\|^2$$
$$= F(\vec{x}_t) - \eta_t\langle \nabla F(\vec{x}_t), \hat{g}_t^{clip} \rangle + \frac{\eta_t^2 L\|\hat{g}_t^{clip}\|^2}{2}$$

Taking expectation of both sides:

$$\mathbb{E}[F(\vec{x}_{t+1})] \leq \mathbb{E}[F(\vec{x}_t)] - \eta_t\mathbb{E}[\langle \nabla F(\vec{x}_t), \hat{g}_t^{clip} \rangle] + \frac{\eta_t^2 L\mathbb{E}[\|\hat{g}_t^{clip}\|^2]}{2}$$
$$\leq \mathbb{E}[F(\vec{x}_t)] - \eta_t\mathbb{E}[\|\nabla F(\vec{x}_t)\|^2] - \eta\mathbb{E}[\langle \nabla F(\vec{x}_t), \hat{\epsilon}_t \rangle] + \frac{\eta_t^2 L\mathbb{E}[\|\hat{g}_t^{clip}\|^2]}{2}$$

Using Young's inequality:

$$\leq \mathbb{E}[F(\vec{x}_t)] - \eta_t\mathbb{E}[\|\nabla F(\vec{x}_t)\|^2] + \frac{\eta_t}{2}\mathbb{E}[\|\nabla F(\vec{x}_t)\|^2] + \frac{\eta_t}{2}\mathbb{E}[\|\hat{\epsilon}_t\|^2] + \frac{\eta_t^2 L\mathbb{E}[\|\hat{g}_t^{clip}\|^2]}{2}$$
$$\leq \mathbb{E}[F(\vec{x}_t)] - \frac{\eta_t}{2}\mathbb{E}[\|\nabla F(\vec{x}_t)\|^2] + \frac{\eta_t}{2}\mathbb{E}[\|\hat{\epsilon}_t\|^2] + \frac{\eta_t^2 L\mathbb{E}[\|\nabla F(\vec{x}_t) + \hat{\epsilon}_t\|^2]}{2}$$

Using $\|a + b\|^2 \leq 2\|a\|^2 + 2\|b\|^2$:

$$\leq \mathbb{E}[F(\vec{x}_t)] - \frac{\eta_t}{2}\mathbb{E}[\|\nabla F(\vec{x}_t)\|^2] + \frac{\eta_t}{2}\mathbb{E}[\|\hat{\epsilon}_t\|^2] + \frac{\eta_t^2 L\mathbb{E}[2\|\nabla F(\vec{x}_t)\| + 2\|\hat{\epsilon}_t\|^2]}{2}$$

Using $\eta_t \leq \frac{1}{4L}$:

$$\leq \mathbb{E}[F(\vec{x}_t)] - \frac{\eta_t}{2}\,\mathbb{E}[\|\nabla F(\vec{x}_t)\|^2] + \frac{\eta_t}{2}\,\mathbb{E}[\|\hat{\epsilon}_t\|^2] + \frac{\eta_t\,\mathbb{E}[\|\nabla F(\vec{x}_t)\| + \|\hat{\epsilon}_t\|^2]}{4}$$

$$= \mathbb{E}[F(\vec{x}_t)] - \frac{\eta_t}{4}\,\mathbb{E}[\|\nabla F(\vec{x}_t)\|^2] + \frac{3\eta_t}{4}\,\mathbb{E}[\|\hat{\epsilon}_t\|^2]$$

$\square$

**Lemma 3.** *Suppose that $f(\vec{x}, z)$ satisfies (2), (4), (6), and (5). Define:*

$$\hat{\epsilon}_t = \hat{g}_t^{clip} - \nabla F(\vec{x}_t)$$

*Now, for some constant $C$ and $\sigma_H$, set $K = \dfrac{2G^2\rho^2}{-2\sigma_H^2 + \sqrt{4\sigma_H^4 + \frac{\rho^2 G^2}{2}}}$, $\eta_t = \frac{1}{Ct^{1/3}}$ with $C \geq \sqrt{2K}$, and $\alpha_t = 2K\eta_t\eta_{t+1}$. Then we have:*

$$\frac{6}{5K\eta_{t+1}}\,\mathbb{E}[\|\hat{\epsilon}_{t+1}\|^2] - \frac{6}{5K\eta_t}\,\mathbb{E}[\|\hat{\epsilon}_t\|^2] \leq -\frac{3\eta_t}{4}\,\mathbb{E}[\|\hat{\epsilon}_t\|^2] + \frac{\eta_t}{5}\,\mathbb{E}[\|\nabla F(\vec{x}_t)\|^2] + \eta_t^3\left(\frac{24}{5}KG^2 + \frac{16C^6G^2}{25K^2}\right)$$

*Proof.* Let us additionally define:

$$\epsilon_t^G = \nabla f(\vec{x}_t, z_t) - \nabla F(\vec{x}_t)$$
$$\epsilon_t^H = \nabla^2 f(\vec{x}_{t+1}, z_{t+1})(\vec{x}_{t+1} - \vec{x}_t) - \nabla^2 F(\vec{x}_{t+1})(\vec{x}_{t+1} - \vec{x}_t)$$

Note that we have the important properties:

$$\mathbb{E}[\epsilon_t^G] = 0$$
$$\mathbb{E}[\epsilon_t^H] = 0$$

Note however that $\mathbb{E}[\hat{\epsilon}_t] \neq 0$. Further, we have:

$$\mathbb{E}[\|\epsilon_t^G\|^2] = \mathbb{E}[\|\nabla f(\vec{x}_t, z_t)\|^2 - 2\langle \nabla f(\vec{x}_t, z_t), \nabla F(\vec{x}_t)\rangle + \|\nabla F(\vec{x}_t)\|^2]$$
$$\leq G^2 - \mathbb{E}[\|\nabla F(\vec{x}_t)\|^2]$$
$$\leq G^2$$

From (4) we have:

$$\mathbb{E}[\|\epsilon_t^H\|^2] \leq \mathbb{E}[\|\nabla^2 f(\vec{x}_{t+1}, z_{t+1})(\vec{x}_{t+1} - \vec{x}_t) - \nabla^2 F(\vec{x}_{t+1})(\vec{x}_{t+1} - \vec{x}_t)\|^2]$$
$$\leq \sigma_H^2\|\vec{x}_{t+1} - \vec{x}_t\|^2$$

Also,

$$\mathbb{E}[\|\hat{\epsilon}_t\|^2] \leq \mathbb{E}[\|\hat{g}_t^{clip} - \nabla F(\vec{x}_t)\|^2]$$
$$\leq 4G^2$$

Finally, also note that we must have:

$$\|\hat{g}_t^{clip}\| \leq G$$

for all $t$ due to our definition of $\hat{g}_t^{clip}$.
Let us define another quantity:

$$\hat{\epsilon}_{t+1}^{\text{noclip}} = \hat{g}_{t+1} - \nabla F(\vec{x}_{t+1})$$

Now, we derive a recursive formula for $\hat{\epsilon}_{t+1}^{\text{noclip}}$ in terms of $\hat{\epsilon}_t$:

$$\hat{\epsilon}_{t+1}^{\text{noclip}} = \hat{g}_{t+1} - \nabla F(\vec{x}_{t+1})$$
$$= (1 - \alpha_t)(\hat{g}_t^{clip} + \nabla^2 f(\vec{x}_{t+1}, z_{t+1})(\vec{x}_{t+1} - \vec{x}_t)) + \alpha_t \nabla f(\vec{x}_{t+1}, z_{t+1}) - \nabla F(\vec{x}_{t+1})$$
$$= (1 - \alpha_t)(\hat{g}_t^{clip} + \nabla^2 f(\vec{x}_{t+1}, z_{t+1})(\vec{x}_{t+1} - \vec{x}_t) - \nabla F(\vec{x}_{t+1})) + \alpha_t(\nabla f(\vec{x}_{t+1}, z_{t+1}) - \nabla F(\vec{x}_{t+1}))$$
$$= (1 - \alpha_t)(\hat{g}_t^{clip} - \nabla F(\vec{x}_t)) + (1 - \alpha_t)(\nabla^2 f(\vec{x}_{t+1}, z_{t+1}) - \nabla^2 F(\vec{x}_{t+1})(\vec{x}_{t+1} - \vec{x}_t))$$
$$\quad + (1 - \alpha_t)(\nabla F(\vec{x}_t) + \nabla^2 F(\vec{x}_{t+1})(\vec{x}_{t+1} - \vec{x}_t) - \nabla F(\vec{x}_{t+1})) + \alpha_t \epsilon_{t+1}^G$$

Now, let's compare $\|\hat{\epsilon}_{t+1}\|$ and $\|\hat{\epsilon}_{t+1}^{\text{noclip}}\|$. If $\|\hat{g}_{t+1}\| \leq G$ (no clipping),

$$\hat{g}_{t+1} = \hat{g}_{t+1}^{clip}$$
$$\Rightarrow \|\hat{\epsilon}_{t+1}\| = \|\hat{\epsilon}_{t+1}^{\text{noclip}}\| \tag{9}$$

If $\|\hat{g}_{t+1}\| > G$, $\|\hat{g}_{t+1}^{clip}\| = G$. Since $\hat{g}_{t+1}^{clip}$ and $\hat{g}_{t+1}$ are co-linear, $\|\hat{g}_{t+1}\| - \|\hat{g}_{t+1}^{clip}\| = \|\hat{g}_{t+1} - \hat{g}_{t+1}^{clip}\|$. Therefore:

$$(\|\hat{g}_{t+1}\| + \|\hat{g}_{t+1}^{clip}\|)(\|\hat{g}_{t+1}\| - \|\hat{g}_{t+1}^{clip}\|) \geq 2G\|\hat{g}_{t+1} - \hat{g}_{t+1}^{clip}\| \tag{10}$$

Using (6) and applying Cauchy-Schwarz inequality, we have:

$$2G\|\hat{g}_{t+1} - \hat{g}_{t+1}^{clip}\| \geq 2\|\nabla F(\vec{x}_{t+1})\|\|\hat{g}_{t+1} - \hat{g}_{t+1}^{clip}\|$$
$$\geq 2\langle \hat{g}_{t+1} - \hat{g}_{t+1}^{clip}, \nabla F(\vec{x}_{t+1})\rangle \tag{11}$$

Combining (10) and (11):

$$(\|\hat{g}_{t+1}\| + \|\hat{g}_{t+1}^{clip}\|)(\|\hat{g}_{t+1}\| - \|\hat{g}_{t+1}^{clip}\|) \geq 2\langle \hat{g}_{t+1} - \hat{g}_{t+1}^{clip}, \nabla F(\vec{x}_{t+1})\rangle$$
$$\|\hat{g}_{t+1}\|^2 - \|\hat{g}_{t+1}^{clip}\|^2 \geq 2\langle \hat{g}_{t+1}, \nabla F(\vec{x}_{t+1})\rangle - 2\langle \hat{g}_{t+1}^{clip}, \nabla F(\vec{x}_{t+1})\rangle$$
$$\|\hat{g}_{t+1}\|^2 - 2\langle \hat{g}_{t+1}, \nabla F(\vec{x}_{t+1})\rangle \geq \|\hat{g}_{t+1}^{clip}\|^2 - 2\langle \hat{g}_{t+1}^{clip}, \nabla F(\vec{x}_{t+1})\rangle$$
$$\|\hat{g}_{t+1}\|^2 - 2\langle \hat{g}_{t+1}, \nabla F(\vec{x}_{t+1})\rangle + \|\nabla F(\vec{x}_{t+1})\|^2 \geq \|\hat{g}_{t+1}^{clip}\|^2 - 2\langle \hat{g}_{t+1}^{clip}, \nabla F(\vec{x}_{t+1})\rangle + \|\nabla F(\vec{x}_{t+1})\|^2$$
$$\|\hat{g}_{t+1} - \nabla F(\vec{x}_{t+1})\|^2 \geq \|\hat{g}_{t+1}^{clip} - \nabla F(\vec{x}_{t+1})\|^2$$
$$\|\hat{\epsilon}_{t+1}^{\text{noclip}}\|^2 \geq \|\hat{\epsilon}_{t+1}\|^2$$
$$\|\hat{\epsilon}_{t+1}^{\text{noclip}}\| \geq \|\hat{\epsilon}_{t+1}\| \tag{12}$$

From relation (9) and (12):

$$\|\hat{\epsilon}_{t+1}\| \leq \|\hat{\epsilon}_{t+1}^{\text{noclip}}\| \tag{13}$$

We have:
$$\hat{\epsilon}_{t+1}^{\text{noclip}} = (1 - \alpha_t)(\hat{g}_t^{clip} - \nabla F(\vec{x}_t)) + (1 - \alpha_t)(\nabla^2 f(\vec{x}_{t+1}, z_{t+1}) - \nabla^2 F(\vec{x}_{t+1})(\vec{x}_{t+1} - \vec{x}_t))$$
$$+ (1 - \alpha_t)(\nabla F(\vec{x}_t) + \nabla^2 F(\vec{x}_{t+1})(\vec{x}_{t+1} - \vec{x}_t) - \nabla F(\vec{x}_{t+1})) + \alpha_t \epsilon_{t+1}^G \tag{14}$$

Let:
$$\delta_t = \nabla F(\vec{x}_t) + \nabla^2 F(\vec{x}_{t+1})(\vec{x}_{t+1} - \vec{x}_t) - \nabla F(\vec{x}_{t+1})$$
$$\|\delta_t\|^2 \leq \frac{\rho^2}{4}\|\vec{x}_{t+1} - \vec{x}_t\|^4 \tag{15}$$

Equation (14) becomes:
$$\hat{\epsilon}_{t+1}^{\text{noclip}} = (1 - \alpha_t)\hat{\epsilon}_t + (1 - \alpha_t)\epsilon_t^H + (1 - \alpha_t)\delta_t + \alpha_t \epsilon_{t+1}^G$$

Now, remember that we are actually interested in $\mathbb{E}[\|\hat{\epsilon}_t\|^2]$, so let us take the norm-squared of both sides in the above and use relation (13):

$$\mathbb{E}[\|\hat{\epsilon}_{t+1}\|^2] \leq (1 - \alpha_t)^2 \mathbb{E}[\|\hat{\epsilon}_t\|^2] + (1 - \alpha_t)^2 \mathbb{E}[\|\epsilon_t^H\|^2] + (1 - \alpha_t)^2 \mathbb{E}[\|\delta_t\|^2] + \alpha_t^2 \mathbb{E}[\|\epsilon_{t+1}^G\|^2] + 2(1 - \alpha_t)^2 \mathbb{E}[\langle \hat{\epsilon}_t, \delta_t\rangle]$$

Applying Young's inequality, for any $\lambda$ we have:

$$\langle \hat{\epsilon}_t, \delta_t\rangle \leq \frac{\lambda\|\hat{\epsilon}_t\|^2}{2} + \frac{\|\delta_t\|^2}{2\lambda} \tag{16}$$

Using (15) and (16), we have:

$$\mathbb{E}[\|\hat{\epsilon}_{t+1}\|^2] \leq (1 - \alpha_t)^2 \mathbb{E}[\|\hat{\epsilon}_t\|^2] + (1 - \alpha_t)^2 \sigma_H^2 \mathbb{E}[\|\vec{x}_{t+1} - \vec{x}_t\|^2] + (1 - \alpha_t)^2 \frac{\rho^2}{4} \mathbb{E}[\|\vec{x}_{t+1} - \vec{x}_t\|^4] + \alpha_t^2 G^2$$
$$+ (1 - \alpha_t)^2 (\lambda \mathbb{E}[\|\hat{\epsilon}_t\|^2] + \frac{\mathbb{E}[\|\delta_t\|^2]}{\lambda})$$
$$\leq (1 - \alpha_t)^2 \mathbb{E}[\|\hat{\epsilon}_t\|^2] + (1 - \alpha_t)^2 \sigma_H^2 \mathbb{E}[\|\vec{x}_{t+1} - \vec{x}_t\|^2] + (1 - \alpha_t)^2 \frac{\rho^2}{4} \mathbb{E}[\|\vec{x}_{t+1} - \vec{x}_t\|^4] + \alpha_t^2 G^2$$
$$+ (1 - \alpha_t)^2 (\lambda \mathbb{E}[\|\hat{\epsilon}_t\|^2]) + (1 - \alpha_t)^2 \frac{\rho^2}{4\lambda} \mathbb{E}[\|\vec{x}_{t+1} - \vec{x}_t\|^4]$$

Next, we observe:

$$\|\vec{x}_t - \vec{x}_{t+1}\| \leq \eta_t \|\hat{g}_t^{clip}\|$$
$$\leq \eta_t(\|\nabla F(\vec{x}_t)\| + \|\hat{\epsilon}_t\|)$$

and:

$$\|\hat{g}_t^{clip}\|^4 \leq G^2 \|\hat{g}_t^{clip}\|^2$$

So plugging this back in yields:

$$\mathbb{E}[\|\hat{\epsilon}_{t+1}\|^2] \leq (1-\alpha_t)^2 \,\mathbb{E}[\|\hat{\epsilon}_t\|^2] + (1-\alpha_t)^2 \sigma_H^2 \eta_t^2 \,\mathbb{E}(\|\nabla F(\vec{x}_t)\|^2 + 2\langle\|\nabla F(\vec{x}_t)\|, \|\hat{\epsilon}_t\|\rangle + \|\hat{\epsilon}_t\|^2)$$

$$+ \alpha_t^2 G^2 + (1-\alpha_t)^2 \frac{\rho^2}{4}\eta_t^4 G^2 \,\mathbb{E}[\|\nabla F(\vec{x}_t)\|^2 + 2\langle\|\nabla F(\vec{x}_t)\|, \|\hat{\epsilon}_t\|\rangle + \|\hat{\epsilon}_t\|^2] + (1-\alpha_t)^2(\lambda\,\mathbb{E}[\|\hat{\epsilon}_t\|^2])$$

$$+ (1-\alpha_t)^2 \frac{\rho^2}{4\lambda}\eta_t^4 G^2 \,\mathbb{E}[]\|\nabla F(\vec{x}_t)\|^2 + 2\langle\|\nabla F(\vec{x}_t)\|, \|\hat{\epsilon}_t\|\rangle + \|\hat{\epsilon}_t\|^2]$$

Again applying Young's Inequality with $\lambda = 1$:

$$\mathbb{E}[\|\hat{\epsilon}_{t+1}\|^2] \leq (1-\alpha_t)^2 \,\mathbb{E}[\|\hat{\epsilon}_t\|^2] + (1-\alpha_t)^2 \sigma_H^2 \eta_t^2 \,\mathbb{E}[2\|\nabla F(\vec{x}_t)\|^2 + 2\|\hat{\epsilon}_t\|^2]$$

$$+ (1-\alpha_t)^2 \frac{\rho^2}{4}\eta_t^4 G^2 \,\mathbb{E}[2\|\nabla F(\vec{x}_t)\|^2 + 2\|\hat{\epsilon}_t\|^2] + \alpha_t^2 G^2$$

$$+ (1-\alpha_t)^2(\lambda\,\mathbb{E}[\|\hat{\epsilon}_t\|^2]) + (1-\alpha_t)^2 \frac{\rho^2}{4\lambda}\eta_t^4 G^2 \,\mathbb{E}(2\|\nabla F(\vec{x}_t)\|^2 + 2\|\hat{\epsilon}_t\|^2)$$

Since $(1-\alpha_t)^2 \leq 1$:

$$\mathbb{E}[\|\hat{\epsilon}_{t+1}\|^2] \leq (1-\alpha_t)^2 \,\mathbb{E}[\|\hat{\epsilon}_t\|^2] + 2\sigma_H^2 \eta_t^2 \,\mathbb{E}[\|\nabla F(\vec{x}_t)\|^2 + \|\hat{\epsilon}_t\|^2] + \alpha_t^2 G^2 + \frac{\rho^2}{2}\eta_t^4 G^2 \,\mathbb{E}[\|\nabla F(x_t)\|^2$$

$$+ \|\hat{\epsilon}_t\|^2] + (\lambda\,\mathbb{E}[\|\hat{\epsilon}_t\|^2]) + \frac{\rho^2}{2\lambda}\eta_t^4 G^2 \,\mathbb{E}[\|\nabla F(\vec{x}_t)\|^2 + \|\hat{\epsilon}_t\|^2]$$

$$\leq \mathbb{E}[\|\hat{\epsilon}_t\|^2]\left[(1-\alpha_t)^2 + 2\sigma_H^2 \eta_t^2 + \frac{\rho^2}{2}\eta_t^4 G^2 + \lambda + \frac{\rho^2}{2\lambda}\eta_t^4 G^2\right] + \alpha_t^2 G^2$$

$$+ \quad \mathbb{E}[\|\nabla F(\vec{x}_t)\|^2]\left[\eta_t^4\left(\frac{\rho^2}{2}G^2 + \frac{\rho^2}{2\lambda}G^2\right) + 2\sigma_H^2 \eta_t^2\right]$$

Now, we will choose parameters in such a way as to ensure:

$$(1-\alpha_t)^2 + 2\sigma_H^2 \eta_t^2 + \frac{\rho^2}{2}\eta_t^4 G^2 + \lambda + \frac{\rho^2}{2\lambda}\eta_t^4 G^2 \leq 1 - \frac{5}{12}\alpha_t \; (*)$$

To this end, let

$$\alpha_t \leq 1$$

and

$$\lambda = \frac{\alpha_t}{2}$$

For (*) to be satisfied:

$$\eta_t^4\left(\frac{\rho^2}{2}G^2 + \frac{\rho^2}{2\lambda}G^2\right) + 2\sigma_H^2 \eta_t^2 - \frac{\alpha_t}{12} \leq 0$$

Solving the quadratic equation, we get:

$$\eta_t^2 \leq \frac{-2\sigma_H^2 + \sqrt{4\sigma_H^4 + \frac{\rho^2 G^2 \alpha_t}{6} + \frac{\rho^2 G^2}{3}}}{G^2\rho^2 + \frac{2G^2\rho^2}{\alpha_t}}$$

$$\leq \frac{-2\sigma_H^2 + \sqrt{4\sigma_H^4 + \frac{\rho^2 G^2}{6} + \frac{\rho^2 G^2}{3}}}{G^2\rho^2 + \frac{2G^2\rho^2}{\alpha_t}}$$

$$\leq \frac{-2\sigma_H^2 + \sqrt{4\sigma_H^4 + \frac{\rho^2 G^2}{2}}}{G^2\rho^2} \frac{\alpha_t}{2}$$

Let $K = \frac{2G^2\rho^2}{-2\sigma_H^2 + \sqrt{4\sigma_H^4 + \frac{\rho^2 G^2}{2}}}$, and suppose that:

$$\eta_t^2 K \leq \alpha_t \tag{17}$$

So overall we get:

$$\mathbb{E}[\|\hat{\epsilon}_{t+1}\|^2] \leq \left(1 - \frac{5}{12}\alpha_t\right)\mathbb{E}[\|\hat{\epsilon}_t\|^2] + \frac{1}{12}\alpha_t\,\mathbb{E}[\|\nabla F(\vec{x}_t)\|^2] + \alpha_t^2 G^2$$

$$\frac{12\eta_t}{5\alpha_t}\mathbb{E}[\|\hat{\epsilon}_{t+1}\|^2 - \|\hat{\epsilon}_t\|^2] \leq -\eta_t\,\mathbb{E}[\|\hat{\epsilon}_t\|^2] + \frac{\eta_t}{5}\mathbb{E}[\|\nabla F(\vec{x}_t)\|^2] + \frac{12}{5}\eta_t\alpha_t G^2$$

Pick $\alpha_t = 2K\eta_t\eta_{t+1}$ and $\eta_t = \frac{1}{Ct^{1/3}}$ ($\frac{\eta_t}{\eta_{t+1}} < 2$ so $\alpha_t$ satisfied (17)):

$$\frac{6}{5K\eta_{t+1}}\mathbb{E}[\|\hat{\epsilon_{t+1}}\|^2 - \|\hat{\epsilon}_t\|^2] \leq -\eta_t\,\mathbb{E}[\|\hat{\epsilon}_t\|^2] + \frac{\eta_t}{5}\mathbb{E}[\|\nabla F(\vec{x}_t)\|^2] + \frac{24}{5}\eta_t^3 KG^2$$

Unfortunately, the coefficient on $\mathbb{E}[\|\hat{\epsilon}_t\|^2]$ above is wrong - it has $\eta_{t+1}$ instead of $\eta_t$. Let's correct that:

$$\frac{6}{5K\eta_{t+1}}\mathbb{E}[\|\hat{\epsilon_{t+1}}\|^2] - \frac{6}{5K\eta_t}\mathbb{E}[\|\hat{\epsilon}_t\|^2] \leq \frac{6}{5K}\left(\frac{1}{\eta_{t+1}} - \frac{1}{\eta_t}\right)\mathbb{E}[\|\hat{\epsilon}_t\|^2] - \eta_t\,\mathbb{E}[\|\hat{\epsilon}_t\|^2] + \frac{\eta_t}{5}\mathbb{E}\,\|\nabla F(\vec{x}_t)\|^2] + \frac{24}{5}\eta_t^3 KG^2$$

So, we need to understand $\frac{1}{\eta_{t+1}} - \frac{1}{\eta_t}$:

$$\frac{1}{\eta_{t+1}} - \frac{1}{\eta_t} = C((t+1)^{1/3} - t^{1/3})$$

$$\leq \frac{C}{3t^{2/3}}$$

$$\leq \frac{C^3\eta_t^2}{3}$$

Now use Young's Inequality ($ab \leq \frac{a^2\lambda}{2} + \frac{b^2}{2\lambda}$) with $a = \sqrt{\eta_t}$ and $b = \eta_t^{3/2}$:

$$\leq \frac{C^3\lambda\eta_t}{6} + \frac{C^3\eta_t^3}{6\lambda}$$

Thus for any $\lambda$ we have:

$$\frac{6}{5K\eta_{t+1}}\mathbb{E}[\|\hat{\epsilon}_{t+1}\|^2] - \frac{6}{5K\eta_t}\mathbb{E}[\|\hat{\epsilon}_t\|^2] \leq \frac{6}{5K}\left(\frac{\lambda C^3\eta_t}{6} + \frac{C^3\eta_t^3}{6\lambda}\right)\mathbb{E}[\|\hat{\epsilon}_t\|^2] - \eta_t\,\mathbb{E}[\|\hat{\epsilon}_t\|^2]$$
$$+ \frac{\eta_t}{5}\mathbb{E}[\|\nabla F(\vec{x}_t)\|^2] + \frac{24}{5}KG^2\eta_t^3$$
$$= -\eta_t\left(1 - \frac{C^3\lambda}{5K}\right)\mathbb{E}[\|\hat{\epsilon}_t\|^2] + \frac{\eta_t}{5}\mathbb{E}[\|\nabla F(\vec{x}_t)\|^2]$$
$$+ \eta_t^3\left(\frac{24}{5}KG^2 + \frac{C^3\mathbb{E}[\|\hat{\epsilon}_t\|^2]}{5K\lambda}\right)$$

So, let us set $\lambda = \frac{5K}{4C^3}$ and use $\mathbb{E}[\|\hat{\epsilon}_t\|^2] \leq 4G^2$:

$$\frac{6}{5K\eta_{t+1}}\mathbb{E}[\|\hat{\epsilon}_{t+1}\|^2] - \frac{6}{5K\eta_t}\mathbb{E}[\|\hat{\epsilon}_t\|^2] \leq -\frac{3\eta_t}{4}\mathbb{E}[\|\hat{\epsilon}_t\|^2] + \frac{\eta_t}{5}\mathbb{E}[\|\nabla F(\vec{x}_t)\|^2]$$
$$+ \eta_t^3\left(\frac{24}{5}KG^2 + \frac{16C^6G^2}{25K^2}\right)$$

$\square$

## A.5   Proof of section 3

**Lemma 9.** *Define:*

$$\hat{\epsilon}_t = \hat{g}_t - \nabla F(\vec{x}_t)$$

Suppose $\vec{x}_1, \ldots, \vec{x}_T$ is a sequence of iterates defined by $\vec{x}_{t+1} = \vec{x}_t - \frac{\hat{g}_t}{\|\hat{g}_t\|}$ for some arbitrary sequence $\hat{g}_1, \ldots, \hat{g}_T$. Then if $\vec{x}_t$ is chosen uniformly at random from $\vec{x}_1, \ldots, \vec{x}_T$, we have:

$$\mathbb{E}[\|\nabla F(\vec{x}_t)\|] \le \frac{3\Delta}{2\eta T} + \frac{3L\eta}{4} + \frac{3}{T}\sum_{t=1}^{T} \|\hat{\epsilon}_t\|$$

*Proof.* Assuming (2) holds, with $\vec{x} = \vec{x}_t$ and $\delta = \vec{x}_{t+1} - \vec{x}_t = \eta\hat{g}_t$, we have:

$$F(\vec{x}_{t+1}) \le F(\vec{x}_t) - \eta\langle\nabla F(\vec{x}_t), \frac{\hat{g}_t}{\|\hat{g}_t\|}\rangle + \frac{L\eta^2}{2} \qquad (18)$$

Let us analyze the inner product term via some case-work: Suppose $\|\hat{\epsilon}_t\| \le \frac{1}{2}\|\nabla F(\vec{x}_t)\|$. Then we have $\|\nabla F(\vec{x}_t) + \hat{\epsilon}_t\| \le \frac{3}{2}\|\nabla F(\vec{x}_t)\|$ so that:

$$
\begin{aligned}
-\langle\nabla F(\vec{x}_t), \frac{\hat{g}_t}{\|\hat{g}_t\|}\rangle &= -\langle\nabla F(\vec{x}_t), \frac{\nabla F(\vec{x}_t) + \hat{\epsilon}_t}{\|\nabla F(\vec{x}_t) + \hat{\epsilon}_t\|}\rangle \\
&\le \frac{-\|\nabla F(\vec{x}_t)\|^2}{\|\nabla F(\vec{x}_t) + \hat{\epsilon}_t\|} + \frac{\|\nabla F(\vec{x}_t)\|\|\hat{\epsilon}_t\|}{\|\nabla F(\vec{x}_t) + \hat{\epsilon}_t\|} \\
&\le -\frac{2}{3}\|\nabla F(\vec{x}_t)\| + 2\|\hat{\epsilon}_t\|
\end{aligned}
$$

On the other hand, if $\|\hat{\epsilon}_t\| > \frac{1}{2}\|\nabla F(\vec{x}_t)\|$, then we have:

$$
\begin{aligned}
-\langle\nabla F(\vec{x}_t), \frac{\hat{g}_t}{\|\hat{g}_t\|}\rangle &\le 0 \\
&\le -\frac{2}{3}\|\nabla F(\vec{x}_t)\| + \frac{2}{3}\|\nabla F(\vec{x}_t)\| \\
&\le -\frac{2}{3}\|\nabla F(\vec{x}_t)\| + \frac{4}{3}\|\hat{\epsilon}_t\|
\end{aligned}
$$

So either way, we have $-\langle\nabla F(\vec{x}_t), \frac{\hat{g}_t}{\|\hat{g}_t\|}\rangle \le -\frac{2}{3}\|\nabla F(\vec{x}_t)\| + 2\|\hat{\epsilon}_t\|$. Now sum (18) over t and rearrange to obtain:

$$
\begin{aligned}
\mathbb{E}[\|\nabla F(\vec{x}_t)\|] &\le \frac{3(F(\vec{x}_1) - F(\vec{x}_{T+1}))}{2\eta} + \frac{3L\eta T}{4} + 3\sum_{t=1}^{T}\|\hat{\epsilon}_t\| \\
&\le \frac{3\Delta}{2\eta} + \frac{3L\eta T}{4} + 3\sum_{t=1}^{T}\|\hat{\epsilon}_t\|
\end{aligned}
$$

Finally, observe that since $\vec{x}_t$ is chosen uniformly at random from $\vec{x}_1, ..., \vec{x}_T$, we have $\mathbb{E}\|\nabla F(\vec{x}_t)\| = \frac{1}{T}\sum_{t=1}^{T}\|\nabla F(\vec{x}_t)\|$ to conclude the results. $\qquad\square$

**Theorem 4.** *Assuming (1), (2), (3), (4), and (5) hold (but* not *assuming (6)), with* $\alpha = \min\{\max\{\frac{1}{T^{2/3}}, \frac{\Delta^{4/5}\rho^{2/5}}{T^{4/5}\sigma_G^{6/5}}, \frac{(2\Delta\sigma_H)^{2/3}}{T^{2/3}\sigma_G^{4/3}}\}, 1\}$ *and* $\eta = \min\{\frac{\sqrt{2\Delta}\alpha^{1/4}}{\sqrt{T(L\sqrt{\alpha}+4\sigma_H)}}, \frac{(\Delta\alpha)^{1/3}}{(\rho T)^{1/3}}\}$, *Algorithm 2 guarantees*

$$
\mathbb{E}[\|\nabla F(\vec{x}_t)\|] \le \frac{6\sigma_G + 54^{1/3}(\Delta\sigma_H)^{1/3}\sigma_G^{1/3}}{T^{1/3}} + \frac{6\sigma_G^{2/5}\Delta^{2/5}\rho^{1/5}}{T^{2/5}} + \frac{\sqrt{9\Delta L} + \sqrt{72\Delta\sigma_H}}{\sqrt{T}} + \frac{6\Delta^{2/3}\rho^{1/3}}{T^{2/3}}
$$
$$
+ \frac{\sqrt{18\Delta\sigma_H}}{T^{3/2}} + \frac{3\Delta^{2/3}\rho^{1/3}}{T^{5/3}}
$$

*In words, Algorithm 2 achieves* $O(1/T^{1/3})$ *with large* $\sigma_H$ *and* $\sigma_G$, *and achieves* $O(1/\sqrt{T})$ *in noiseless case, without requiring a Lipschitz bound on the objective.*

*Proof.* Let us write a recursive expression for $\hat{\epsilon}_t$:

$$
\begin{aligned}
\hat{\epsilon}_t &= \hat{g}_t - \nabla F(\vec{x}_t) \\
&= (1-\alpha)(\hat{g}_{t-1} + \nabla^2 f(\vec{x}_t, z_t)(\vec{x}_t - \vec{x}_{t-1})) + \alpha\nabla f(\vec{x}_t, z_t) - \nabla F(\vec{x}_t) \\
&= (1-\alpha)(\hat{g}_{t-1} + \nabla^2 f(\vec{x}_t, z_t)(\vec{x}_t - \vec{x}_{t-1}) - \nabla F(\vec{x}_t)) + \alpha(\nabla f(\vec{x}_t, z_t) - \nabla F(\vec{x}_t))
\end{aligned}
$$

Let us define $v_t = \nabla f(\vec{x}_t, z_t) - \nabla F(\vec{x}_t)$ and $w_t = \frac{\nabla^2 f(\vec{x}_t, z_t)(\vec{x}_t - \vec{x}_{t-1}) - \nabla^2 F(\vec{x}_t)(\vec{x}_t - \vec{x}_{t-1})}{\|\vec{x}_t - \vec{x}_{t-1}\|}$. Note that:

$$w_t = \frac{\nabla^2 f(\vec{x}_t, z_t)(\vec{x}_t - \vec{x}_{t-1}) - \nabla^2 F(\vec{x}_t)(\vec{x}_t - \vec{x}_{t-1})}{\|\vec{x}_t - \vec{x}_{t-1}\|}$$
$$= \frac{\nabla^2 f(\vec{x}_t, z_t)(\vec{x}_t - \vec{x}_{t-1}) - \nabla^2 F(\vec{x}_t)(\vec{x}_t - \vec{x}_{t-1})}{\eta}$$

Finally, define $\delta_t = \nabla F(\vec{x}_{t-1}) + \nabla^2 F(\vec{x}_t)(\vec{x}_t - \vec{x}_{t-1}) - \nabla F(\vec{x}_t)$. Since F is $\rho$-second-order smooth, we must have $\|\delta_t\| \le \frac{\rho}{2}\|\vec{x}_t - \vec{x}_{t-1}\|^2 = \frac{\rho}{2}\eta^2$. Now we write:

$$\hat{\epsilon}_t = (1-\alpha)[\hat{g}_{t-1} + \nabla^2 f(\vec{x}_t, z_t)(\vec{x}_t - \vec{x}_{t-1}) - \nabla F(\vec{x}_t)] + \alpha(\nabla f(\vec{x}_t, z_t) - \nabla F(\vec{x}_t))$$
$$= (1-\alpha)[\hat{g}_{t-1} - \nabla F(\vec{x}_{t-1}) + \|\vec{x}_t - \vec{x}_{t-1}\|w_t + \nabla F(\vec{x}_{t-1}) + \nabla^2 F(\vec{x}_t, z_t)(\vec{x}_t - \vec{x}_{t-1}) - \nabla F(\vec{x}_t)] + \alpha v_t$$
$$= (1-\alpha)(\hat{\epsilon}_{t-1} + \|\vec{x}_t - \vec{x}_{t-1}\|w_t + \delta_t) + \alpha v_t$$

Now unroll this recursive expression:

$$\hat{\epsilon}_t = (1-\alpha)^{t-1}\hat{\epsilon}_1 + \sum_{\tau=0}^{t-1}(1-\alpha)^{\tau+1}(\|\vec{x}_t - \vec{x}_{t-1}\|w_{t-\tau} + \delta_{t-\tau}) + \alpha(1-\alpha)^\tau y_{t-\tau}$$

Observe that $\hat{\epsilon}_1 = v_1$ and apply triangle inequality:

$$\hat{\epsilon}_t \le (1-\alpha)^{t-1}\|v_1\| + \frac{\eta^2\rho}{2}\sum_{\tau=0}^{t-1}(1-\alpha)^{\tau+1} + \eta\|\sum_{\tau=0}^{t-1}(1-\alpha)^{\tau+1}w_{t-\tau}\| + \alpha\|\sum_{\tau=0}^{t-1}(1-\alpha)^\tau v_{t-\tau}\|$$

Take expectation of the expression:

$$\mathbb{E}[\|\hat{\epsilon}_t\|] \le (1-\alpha)^{t-1}\sigma_G + \frac{\eta^2\rho}{2}\sum_{\tau=0}^{t-1}(1-\alpha)^{\tau+1} + \eta\sigma_H\sqrt{\sum_{\tau=0}^{t-1}(1-\alpha)^{2\tau+2}} + \sigma_G\alpha\sqrt{\sum_{\tau=0}^{t-1}(1-\alpha)^{2\tau}}$$

All the sums can be upper bounded by $\sum_{\tau=0}^\infty(1-\alpha)^\tau = \frac{1}{\alpha}$:

$$\le (1-\alpha)^{t-1}\sigma_G + \frac{\eta^2\rho}{2\alpha} + \frac{\eta\sigma_G}{\sqrt\alpha} + \sigma_G\sqrt\alpha$$

Next, sum over t:

$$\sum_{t=1}^T \mathbb{E}[\|\hat{\epsilon}_t\|] \le \frac{\sigma_G}{\alpha} + \frac{\eta^2\rho T}{2\alpha} + \frac{\eta\sigma_H T}{\sqrt\alpha} + \sigma_G\sqrt\alpha T$$

Applying Lemma 9:

$$\mathbb{E}[\|\nabla F(\vec{x}_t)\|] \le \frac{3\Delta}{2\eta T} + \frac{3L\eta}{4} + \frac{3}{T}\sum_{t=1}^T\|\hat{\epsilon}_t\|$$
$$\le \frac{3\Delta}{2\eta T} + \frac{3L\eta}{4} + \frac{3\sigma_G}{T\alpha} + \frac{3\eta^2\rho}{2\alpha} + \frac{3\eta\sigma_H}{\sqrt\alpha} + 3\sigma_G\sqrt\alpha$$

Now, with $\alpha = \min\{\max\{\frac{1}{T^{2/3}}, \frac{\Delta^{4/5}\rho^{2/5}}{T^{4/5}\sigma_G^{6/5}}, \frac{(2\Delta\sigma_H)^{2/3}}{T^{2/3}\sigma_G^{4/3}}\}, 1\}$ and $\eta = \min\{\frac{\sqrt{2\Delta}\alpha^{1/4}}{\sqrt{T}(L\sqrt\alpha + 4\sigma_H)}, \frac{(\Delta\alpha)^{1/3}}{(\rho T)^{1/3}}\}$, use Lemma 8 in the appendix to finish the proof. $\qquad\square$

## A.6  Proof of section 4

**Lemma 10.** *Define:*

$$\hat{\epsilon}_t = \hat{g}_t^{clip} - \nabla F(\vec{x}_t)$$

*Now, for some constant $\sigma_H$, set $K = \frac{2G^2\rho^2}{-4\sigma_H^2 + \sqrt{16\sigma_H^4 + \frac{\rho^2 G^2}{2}}}$, $\eta_t = \frac{c}{(w + \sum_{i=1}^{t-2} G_i^2)^{1/3}}$ with $c \leq \frac{2G^{2/3}}{\sqrt{K}}$ and $w = \max\{(4Lc)^3, 3G^2\}$, $\alpha_t = 2K\eta_t\eta_{t+1}$. Then we have:*

$$\sum_{t=1}^{T} \mathbb{E}\left[\frac{6}{5K\eta_{t+1}}\|\hat\epsilon_{t+1}\|^2 - \frac{6}{5K\eta_t}\|\hat\epsilon_t\|^2\right] \leq \sum_{t=1}^{T} E\left[-\frac{3\eta_t}{4}\|\hat\epsilon_t\|^2 + \frac{\eta_t}{5}\|\nabla F(\vec{x}_t)\|^2\right]$$

$$+ \frac{48}{5}Kc^2\ln(T+1) + \frac{16G^4}{25K^2c^3}\ln T$$

*Proof.* Similar to Lemma 3, let us define:

$$\epsilon_t^G = \nabla f(\vec{x}_t, z_t) - \nabla F(\vec{x}_t)$$
$$\epsilon_t^H = \nabla^2 f(\vec{x}_{t+1}, z_{t+1})(\vec{x}_{t+1} - \vec{x}_t) - \nabla^2 F(\vec{x}_{t+1})(\vec{x}_{t+1} - \vec{x}_t)$$

Note that we have the important properties:

$$\mathbb{E}[\epsilon_t^G | z_1, z_2, ..., z_{t-1}] = 0$$
$$\mathbb{E}[\epsilon_t^H | z_1, z_2, ..., z_t] = 0$$

Further, we have:

$$\mathbb{E}[\|\epsilon_t^G\|^2] \leq \mathbb{E}[G_t^2]$$

And:

$$\mathbb{E}[\|\epsilon_t^H\|^2] \leq \mathbb{E}[\|\nabla^2 f(\vec{x}_{t+1}, z_{t+1})(\vec{x}_{t+1} - \vec{x}_t) - \nabla^2 F(\vec{x}_{t+1})(\vec{x}_{t+1} - \vec{x}_t)\|^2]$$
$$\leq \sigma_H^2 \|\vec{x}_{t+1} - \vec{x}_t\|^2$$

Also,

$$\mathbb{E}[\|\hat\epsilon_t\|^2] \leq \mathbb{E}[\|\hat{g}_t^{clip} - \nabla F(\vec{x}_t)\|^2]$$
$$\leq 4G^2$$

Finally, also note that we must have:

$$\|\hat{g}_t^{clip}\| \leq G$$

for all $t$ due to our definition of $\hat{g}_t^{clip}$.
Let us define another quantity:

$$\hat\epsilon_{t+1}^{\text{noclip}} = \hat{g}_{t+1} - \nabla F(\vec{x}_{t+1})$$

Now, we derive a recursive formula for $\hat\epsilon_{t+1}^{\text{noclip}}$ in terms of $\hat\epsilon_t$:

$$\hat\epsilon_{t+1}^{\text{noclip}} = \hat{g}_{t+1} - \nabla F(\vec{x}_{t+1})$$
$$= (1 - \alpha_t)(\hat{g}_t^{clip} + \nabla^2 f(\vec{x}_{t+1}, z_{t+1})(\vec{x}_{t+1} - \vec{x}_t)) + \alpha_t \nabla f(\vec{x}_{t+1}, z_{t+1}) - \nabla F(\vec{x}_{t+1})$$
$$= (1 - \alpha_t)(\hat{g}_{tclip} + \nabla^2 f(\vec{x}_{t+1}, z_{t+1})(\vec{x}_{t+1} - \vec{x}_t) - \nabla F(\vec{x}_{t+1})) + \alpha_t(\nabla f(\vec{x}_{t+1}, z_{t+1}) - \nabla F(\vec{x}_{t+1}))$$
$$= (1 - \alpha_t)(\hat{g}_t^{clip} - \nabla F(\vec{x}_t)) + (1 - \alpha_t)(\nabla^2 f(\vec{x}_{t+1}, z_{t+1}) - \nabla^2 F(\vec{x}_{t+1})(\vec{x}_{t+1} - \vec{x}_t))$$
$$+ (1 - \alpha_t)(\nabla F(\vec{x}_t) + \nabla^2 F(\vec{x}_{t+1})(\vec{x}_{t+1} - \vec{x}_t) - \nabla F(\vec{x}_{t+1})) + \alpha_t \epsilon_{t+1}^G$$

From the analysis of Lemma 3, we have the relation:

$$\|\hat\epsilon_{t+1}\| \leq \|\hat\epsilon_{t+1}^{\text{noclip}}\| \tag{19}$$

And we also have:

$$\hat\epsilon_{t+1}^{\text{noclip}} = (1 - \alpha_t)(\hat{g}_t^{clip} - \nabla F(\vec{x}_t)) + (1 - \alpha_t)(\nabla^2 f(\vec{x}_{t+1}, z_{t+1}) - \nabla^2 F(\vec{x}_{t+1})(\vec{x}_{t+1} - \vec{x}_t)) + (1 - \alpha_t)(\nabla F(\vec{x}_t)$$
$$+ \nabla^2 F(\vec{x}_{t+1})(\vec{x}_{t+1} - \vec{x}_t) - \nabla F(\vec{x}_{t+1})) + \alpha_t \epsilon_{t+1}^G \tag{20}$$

Let:

$$\delta_t = \nabla F(\vec{x}_t) + \nabla^2 F(\vec{x}_{t+1})(\vec{x}_{t+1} - \vec{x}_t) - \nabla F(\vec{x}_{t+1})$$

$$\|\delta_t\|^2 \le \frac{\rho^2}{4}\|\vec{x}_{t+1} - \vec{x}_t\|^4 \tag{21}$$

Equation (20) becomes:

$$\hat{\epsilon}_{t+1}^{\text{noclip}} = (1-\alpha_t)\hat{\epsilon}_t + (1-\alpha_t)\epsilon_t^H + (1-\alpha_t)\delta_t + \alpha_t\epsilon_{t+1}^G$$

Now use relation (19), we have:

$$\|\hat{\epsilon}_{t+1}\|^2 \le \|(1-\alpha_t)\hat{\epsilon}_t + (1-\alpha_t)\epsilon_t^H + (1-\alpha_t)\delta_t + \alpha_t\epsilon_{t+1}^G\|^2$$

Multiply $\frac{12\eta_t}{5\eta_t}$ to both sides and take the expectation of the above equation:

$$\mathbb{E}[\frac{12\eta_t}{5\alpha_t}\|\hat{\epsilon}_{t+1}\|^2] \le \mathbb{E}[\frac{12\eta_t}{5\alpha_t}\|(1-\alpha_t)\hat{\epsilon}_t + (1-\alpha_t)\epsilon_t^H + (1-\alpha_t)\delta_t + \alpha_t\epsilon_{t+1}^G\|^2]$$

Notice that by definition $\frac{\eta_t}{\alpha_t} = \frac{\eta_t}{2K\eta_t\eta_{t+1}} = \frac{1}{2K\eta_{t+1}} = \frac{2K(w+\sum_{i=1}^{t-1}G_i^2)^{1/3}}{c}$ which is independent of the current sample $z_t$. Thus when we take expectation with respect to sample $z_t$, we can consider $\frac{\eta_t}{\alpha_t}$ as a constant. For example, let us analyze $\mathbb{E}\left[\frac{\eta_t}{\alpha_t}\langle\hat{\epsilon}_t, \epsilon_t^H\rangle\right]$:

$$\mathbb{E}_{z_1,...,z_t}\left[\frac{\eta_t}{\alpha_t}\langle\hat{\epsilon}_t, \epsilon_t^H\rangle\right] = \mathbb{E}_{z_1,...,z_{t-1}}\left[\mathbb{E}_{z_t}\left[\frac{\eta_t}{\alpha_t}\langle\hat{\epsilon}_t, \epsilon_t^H\rangle|z_1,...,z_{t-1}\right]\right]$$

$$= \mathbb{E}_{z_1,...,z_{t-1}}\left[\frac{\eta_t}{\alpha_t}\mathbb{E}_{z_t}\left[\langle\hat{\epsilon}_t, \epsilon_t^H\rangle|z_1,...,z_{t-1}\right]\right]$$

Then the cross-terms $E[\langle\hat{\epsilon}_t, \epsilon_t^H\rangle], E[\langle\delta_t, \epsilon_t^H\rangle], E[\langle\epsilon_{t+1}^G, \hat{\epsilon}_t\rangle], E[\langle\epsilon_{t+1}^G, \delta_t\rangle]$ all become zero in expectation. Then:

$$\mathbb{E}[\frac{12\eta_t}{\alpha_t}\|\hat{\epsilon}_{t+1}\|^2] \le \mathbb{E}[\frac{12\eta_t}{5\alpha_t}[(1-\alpha_t)^2(\|\hat{\epsilon}_t\|^2 + \|\epsilon_t^H\|^2 + \|\delta_t\|^2) + \alpha_t^2\|\epsilon_{t+1}^G\|^2 + 2(1-\alpha_t)^2\langle\hat{\epsilon}_t, \delta_t\rangle$$
$$+ 2\langle(1-\alpha_t)\epsilon_t^H, \alpha_t\epsilon_{t+1}^G\rangle]]$$

Applying Young's inequality, for any $\lambda$ we have:

$$\langle\hat{\epsilon}_t, \delta_t\rangle \le \frac{\lambda\|\hat{\epsilon}_t\|^2}{2} + \frac{\|\delta_t\|^2}{2\lambda} \tag{22}$$

and:

$$\langle(1-\alpha_t)\epsilon_t^H, \alpha_t\epsilon_{t+1}^G\rangle \le \frac{(1-\alpha_t)^2\|\epsilon_t^H\|^2}{2} + \frac{\alpha_t^2\|\epsilon_{t+1}^G\|^2}{2} \tag{23}$$

Using (21),(22), and (16), we have:

$$\mathbb{E}[\frac{12\eta_t}{\alpha_t}\|\hat{\epsilon}_{t+1}\|^2] \le \mathbb{E}[\frac{12\eta_t}{\alpha_t}[(1-\alpha_t)^2\|\hat{\epsilon}_t\|^2 + 2(1-\alpha_t)^2\sigma_H^2\|\vec{x}_{t+1} - \vec{x}_t\|^2 + (1-\alpha_t)^2\frac{\rho^2}{4}\|\vec{x}_{t+1} - \vec{x}_t\|^4 + 2\alpha_t^2 G_t^2$$
$$+ (1-\alpha_t)^2(\lambda\|\hat{\epsilon}_t\|^2 + \frac{\|\delta_t\|^2}{\lambda})]]$$

$$\le \mathbb{E}[\frac{12\eta_t}{\alpha_t}[(1-\alpha_t)^2\|\hat{\epsilon}_t\|^2 + 2(1-\alpha_t)^2\sigma_H^2\|\vec{x}_{t+1} - \vec{x}_t\|^2 + (1-\alpha_t)^2\frac{\rho^2}{4}\|\vec{x}_{t+1} - \vec{x}_t\|^4 + 2\alpha_t^2 G_t^2$$
$$+ (1-\alpha_t)^2(\lambda\|\hat{\epsilon}_t\|^2) + (1-\alpha_t)^2\frac{\rho^2}{4\lambda}\|\vec{x}_{t+1} - \vec{x}_t\|^4]]$$

Next, we observe:

$$\|\vec{x}_t - \vec{x}_{t+1}\| \le \eta_t\|\hat{g}_t^{clip}\|$$
$$\le \eta_t(\|\nabla F(\vec{x}_t)\| + \|\hat{\epsilon}_t\|)$$

and:
$$\|\hat{g}_t^{clip}\|^4 \le G^2 \|\hat{g}_t^{clip}\|^2$$

So plugging this back in yields:

$$\mathbb{E}[\frac{12\eta_t}{\alpha_t}\|\hat{\epsilon}_{t+1}\|^2] \le \mathbb{E}[\frac{12\eta_t}{\alpha_t}[(1-\alpha_t)^2[\|\hat{\epsilon}_t\|^2 + 2(1-\alpha_t)^2\sigma_H^2\eta_t^2(\|\nabla F(\vec{x}_t)\|^2 + 2\langle\|\nabla F(\vec{x}_t)\|, \|\hat{\epsilon}_t\|\rangle + \|\hat{\epsilon}_t\|^2) + 2\alpha_t^2 G_t^2$$

$$+ (1-\alpha_t)^2\frac{\rho^2}{4}\eta_t^4 G^2(\|\nabla F(\vec{x}_t)\|^2 + 2\langle\|\nabla F(\vec{x}_t)\|, \|\hat{\epsilon}_t\|\rangle + \|\hat{\epsilon}_t\|^2) + (1-\alpha_t)^2(\lambda\|\hat{\epsilon}_t\|^2)$$

$$+ (1-\alpha_t)^2\frac{\rho^2}{4\lambda}\eta_t^4 G^2(\|\nabla F(\vec{x}_t)\|^2 + 2\langle\|\nabla F(\vec{x}_t)\|, \|\hat{\epsilon}_t\|\rangle + \|\hat{\epsilon}_t\|^2)]]$$

Again applying Young's Inequality with $\lambda = 1$:

$$\mathbb{E}[\frac{12\eta_t}{\alpha_t}\|\hat{\epsilon}_{t+1}\|^2] \le \mathbb{E}[\frac{12\eta_t}{\alpha_t}[(1-\alpha_t)^2\|\hat{\epsilon}_t\|^2] + 2(1-\alpha_t)^2\sigma_H^2\eta_t^2(2\|\nabla F(\vec{x}_t)\|^2 + 2\|\hat{\epsilon}_t\|^2)$$

$$+ (1-\alpha_t)^2\frac{\rho^2}{4}\eta_t^4 G^2(2\|\nabla F(\vec{x}_t)\|^2 + 2\|\hat{\epsilon}_t\|^2)$$

$$+ (1-\alpha_t)^2(\lambda\|\hat{\epsilon}_t\|^2) + (1-\alpha_t)^2\frac{\rho^2}{4\lambda}\eta_t^4 G^2(2\|\nabla F(\vec{x}_t)\|^2 + 2\|\hat{\epsilon}_t\|^2)]] + 2\alpha_t^2 G_t^2$$

Since $(1-\alpha_t)^2 \le 1$:

$$\mathbb{E}[\frac{12\eta_t}{\alpha_t}\|\hat{\epsilon}_{t+1}\|^2] \le \mathbb{E}[\frac{12\eta_t}{\alpha_t}[(1-\alpha_t)^2\|\hat{\epsilon}_t\|^2] + 4\sigma_H^2\eta_t^2(\|\nabla F(\vec{x}_t)\|^2 + \|\hat{\epsilon}_t\|^2) + 2\alpha_t^2 G_t^2$$

$$+ \frac{\rho^2}{2}\eta_t^4 G^2(\|\nabla F(x_t)\|^2 + \|\hat{\epsilon}_t\|^2) + (\lambda\|\hat{\epsilon}_t\|^2) + \frac{\rho^2}{2\lambda}\eta_t^4 G^2(\|\nabla F(\vec{x}_t)\|^2 + \|\hat{\epsilon}_t\|^2)]]$$

$$= E[\frac{12\eta_t}{\alpha_t}[\|\hat{\epsilon}_t\|^2((1-\alpha_t)^2 + 4\sigma_H^2\eta_t^2 + \frac{\rho^2}{2}\eta_t^4 G^2 + \lambda + \frac{\rho^2}{2\lambda}\eta_t^4 G^2)$$

$$+ \|\nabla F(\vec{x}_t)\|^2(4\sigma_H^2\eta_t^2 + \frac{\rho^2}{2}\eta_t^4 G^2 + \lambda + \frac{\rho^2}{2\lambda}\eta_t^4 G^2) + 2\alpha_t^2 G_t^2]]$$

Now we want the coefficient of the error to be something like 1-$O(\alpha_t)$:

$$(1-\alpha_t)^2 + 4\sigma_H^2\eta_t^2 + \frac{\rho^2}{2}\eta_t^4 G^2 + \lambda + \frac{\rho^2}{2\lambda}\eta_t^4 G^2 \le 1 - \frac{5}{12}\alpha_t \; (*)$$

Let
$$\alpha_t \le 1$$
and
$$\lambda = \frac{\alpha_t}{2}$$

For (*) to be satisfied:

$$\eta_t^4(\frac{\rho^2}{2}G^2 + \frac{\rho^2}{2\lambda}G^2) + 4\sigma_H^2\eta_t^2 - \frac{\alpha_t}{12} \le 0$$

Solving the quadratic equation, we get:

$$\eta_t^2 \le \frac{-4\sigma_H^2 + \sqrt{16\sigma_H^4 + \frac{\rho^2 G^2 \alpha_t}{6} + \frac{\rho^2 G^2}{3}}}{G^2\rho^2 + \frac{2G^2\rho^2}{\alpha_t}}$$

$$\le \frac{-4\sigma_H^2 + \sqrt{16\sigma_H^4 + \frac{\rho^2 G^2}{6} + \frac{\rho^2 G^2}{3}}}{G^2\rho^2 + \frac{2G^2\rho^2}{\alpha_t}}$$

$$\le \frac{-4\sigma_H^2 + \sqrt{16\sigma_H^4 + \frac{\rho^2 G^2}{2}}}{G^2\rho^2}\frac{\alpha_t}{2}$$

Let $K = \frac{2G^2\rho^2}{-4\sigma_H^2 + \sqrt{16\sigma_H^4 + \frac{\rho^2 G^2}{2}}}$:

$$\eta_t^2 K \leq \alpha_t \tag{24}$$

So overall we get:

$$\mathbb{E}[\frac{12\eta_t}{5\alpha_t}\|\hat{\epsilon}_{t+1}\|^2] \leq \mathbb{E}[\frac{12\eta_t}{5\alpha_t}[(1 - \frac{5}{12}\alpha_t)\|\hat{\epsilon}_t\|^2 + \frac{1}{12}\alpha_t\|\nabla F(\vec{x}_t)\|^2 + 2\alpha_t^2 G_t^2]]$$

Let $\eta_t = \frac{c}{(w + \sum_{i=1}^{t-2} G_i^2)^{1/3}}$ and $\alpha_t = 2K\eta_t\eta_{t+1}$. Then:

$$\mathbb{E}[\frac{12\eta_t}{5\alpha_t}\|\hat{\epsilon}_{t+1}\|^2] \leq \mathbb{E}\left[\frac{12\eta_t}{5\alpha_t}[(1 - \frac{5}{12}\alpha_t)\|\hat{\epsilon}_t\|^2 + \frac{1}{12}\alpha_t\|\nabla F(\vec{x}_t)\|^2 + 2\alpha_t^2 G_t^2]\right]$$

$$= \mathbb{E}\left[\frac{12\eta_t}{5\alpha_t}\|\hat{\epsilon}_t\|^2 - \eta_t\|\hat{\epsilon}_t\|^2 + \frac{\eta_t}{5}\|\nabla F(\vec{x}_t)\|^2 + \frac{24}{5}\eta_t\alpha_t G_t^2\right]$$

$$\Rightarrow \mathbb{E}[\frac{12\eta_t}{5\alpha_t}\|\hat{\epsilon}_{t+1}\|^2 - \frac{12\eta_t}{5\alpha_t}\|\hat{\epsilon}_t\|^2] \leq \mathbb{E}\left[-\eta_t\|\hat{\epsilon}_t\|^2 + \frac{\eta_t}{5}\|\nabla F(\vec{x}_t)\|^2 + \frac{24}{5}\eta_t\alpha_t G_t^2\right]$$

$$\Leftrightarrow \mathbb{E}[\frac{6}{5K\eta_{t+1}}\|\hat{\epsilon}_{t+1}\|^2 - \frac{6}{5K\eta_{t+1}}\|\hat{\epsilon}_t\|^2] \leq \mathbb{E}\left[-\eta_t\|\hat{\epsilon}_t\|^2 + \frac{\eta_t}{5}\|\nabla F(\vec{x}_t)\|^2 + \frac{48}{5}\eta_t^3 K G_t^2\right]$$

Subtract $E[\frac{6}{5K\eta_t}\|\hat{\epsilon}_t\|^2]$ from both sides:

$$\mathbb{E}[\frac{6}{5K\eta_{t+1}}\|\hat{\epsilon}_{t+1}\|^2 - \frac{6}{5K\eta_t}\|\hat{\epsilon}_t\|^2] \leq \mathbb{E}\left[\frac{6}{5K}(\frac{1}{\eta_{t+1}} - \frac{1}{\eta_t})\|\hat{\epsilon}_t\|^2 - \eta_t\|\hat{\epsilon}_t\|^2 + \frac{\eta_t}{5}\|\nabla F(\vec{x}_t)\|^2 + \frac{48}{5}\eta_t^3 K G_t^2\right]$$

Now, let us analyze $\frac{1}{\eta_{t+1}} - \frac{1}{\eta_t}$:

$$\frac{1}{\eta_{t+1}} - \frac{1}{\eta_t} = \frac{1}{c}\left[(w + \sum_{i=1}^{t-1} G_i^2)^{1/3} - (w + \sum_{i=1}^{t-2} G_i^2)^{1/3}\right]$$

$$\leq \frac{G_{t-1}^2}{3c(w + \sum_{i=1}^{t-2} G_i^2)^{2/3}}$$

$$= \frac{k^2 G_{t-1}^2}{3c^3(w + \sum_{i=1}^{t-2} G_i^2)^{2/3}}$$

$$= \frac{\eta_t^2 G_{t-1}^2}{3c^3}$$

$$\leq \frac{G_{t-1}^2 \lambda \eta_t}{6c^3} + \frac{G_{t-1}^2 \eta_t^3}{6c^3\lambda}$$

Plug in:

$$\mathbb{E}[\frac{6}{5K\eta_{t+1}}\|\hat{\epsilon}_{t+1}\|^2 - \frac{6}{5K\eta_t}\|\hat{\epsilon}_t\|^2] \leq \mathbb{E}\left[\frac{6}{5K}(\frac{G_{t-1}^2\lambda\eta_t}{6c^3} + \frac{G_{t-1}^2\eta_t^3}{6c^3\lambda})\|\hat{\epsilon}_t\|^2 - \eta_t\|\hat{\epsilon}_t\|^2 + \frac{\eta_t}{5}\|\nabla F(\vec{x}_t)\|^2 + \frac{24}{5}\eta_t^3 K G_t^2\right]$$

$$= \mathbb{E}\left[-\eta_t\|\hat{\epsilon}_t\|^2\left(1 - \frac{G_{t-1}^2\lambda}{5Kc^3}\right) + \frac{\eta_t}{5}\|\nabla F(\vec{x}_t)\|^2 + \eta_t^3\left(\frac{48}{5}KG_t^2 + \frac{G_{t-1}^2\|\hat{\epsilon}_t\|^2}{5Kc^3\lambda}\right)\right]$$

Let $\lambda = \frac{5Kc^3}{4G_{t-1}^2}$ and use the fact that $\mathbb{E}[\|\hat{\epsilon}\|^2] \leq 4G^2$:

$$\leq \mathbb{E}\left[-\frac{3\eta_t}{4}\|\hat{\epsilon}_t\|^2 + \frac{\eta_t}{5}\|\nabla F(\vec{x}_t)\|^2 + \eta_t^3\left(\frac{48}{5}KG_t^2 + \frac{16G^2 G_{t-1}^4}{25K^2 c^6}\right)\right]$$

$$\leq \mathbb{E}\left[-\frac{3\eta_t}{4}\|\hat{\epsilon}_t\|^2 + \frac{\eta_t}{5}\|\nabla F(\vec{x}_t)\|^2 + \eta_t^3\left(\frac{48}{5}KG_t^2 + \frac{16G^4 G_{t-1}^2}{25K^2 c^6}\right)\right]$$

Now sum over t:

$$\sum_{t=1}^{T} \mathbb{E}[\frac{6}{5K\eta_{t+1}}\|\hat{\epsilon}_{t+1}\|^2 - \frac{6}{5K\eta_t}\|\hat{\epsilon}_t\|^2] \le \sum_{t=1}^{T} \mathbb{E}\left[-\frac{3\eta_t}{4}\|\hat{\epsilon}_t\|^2 + \frac{\eta_t}{5}\|\nabla F(\vec{x}_t)\|^2 + \eta_t^3\left(\frac{48}{5}KG_t^2 + \frac{16G^4G_{t-1}^2}{25K^2c^6}\right)\right]$$

From Lemma 4 of Cutkosky and Orabona [2019], we have the following:

$$\sum_{t=1}^{T}\frac{a_t}{a_0 + \sum_{i=1}^{t}a_i} \le \ln\left(1 + \frac{\sum_{i=1}^{t}a_i}{a_0}\right)$$

Analyze third term:

$$\sum_{t=1}^{T}\frac{48}{5}K\eta_t^3 G_t^2 = \sum_{t=1}^{T}\frac{48}{5}Kc^3\frac{G_t^2}{w + \sum_{i=1}^{t-2}G_i^2}$$

Now with $w \ge 3G^2 \ge G^2 + G_{t-1}^2 + G_t^2$, we would get:

$$\sum_{t=1}^{T}\frac{48}{5}K\eta_t^3 G_t^2 \le \sum_{t=1}^{T}\frac{48}{5}Kc^3\frac{G_t^2}{G^2 + \sum_{i=1}^{t}G_i^2}$$

$$\le \frac{48}{5}Kc^2\ln\left(1 + \frac{\sum_{i=1}^{T}G_i^2}{G^2}\right)$$

$$\le \frac{48}{5}Kc^2\ln(T+1)$$

Analyze the fourth term:

$$\sum_{t=1}^{T}\frac{16G^4G_{t-1}^2\eta_t^3}{25K^2c^6} = \sum_{t=1}^{T}\frac{16G^4}{25K^2c^3}\frac{G_{t-1}^2}{w + \sum_{i=1}^{t-2}G_i^2}$$

$$\le \sum_{t=1}^{T}\frac{16G^4}{25K^2c^3}\frac{G_{t-1}^2}{2G^2 + \sum_{i=1}^{t-1}G_i^2}$$

$$\le \frac{16G^4}{25K^2c^3}\ln\left(1 + \sum_{i=1}^{T-1}\frac{G_i^2}{G^2}\right)$$

$$\le \frac{16G^4}{25K^2c^3}\ln T$$

Then:

$$\sum_{t=1}^{T}\mathbb{E}[\frac{6}{5K\eta_{t+1}}\|\hat{\epsilon}_{t+1}\|^2 - \frac{6}{5K\eta_t}\|\hat{\epsilon}_t\|^2] \le \sum_{t=1}^{T}E\left[-\frac{3\eta_t}{4}\|\hat{\epsilon}_t\|^2 + \frac{\eta_t}{5}\|\nabla F(\vec{x}_t)\|^2\right] + \frac{48}{5}Kc^2\ln(T+1) + \frac{16G^4}{25K^2c^3}\ln T$$

$\square$

**Theorem 5.** *With* $K = \frac{2G^2\rho^2}{-4\sigma_H^2 + \sqrt{16\sigma_H^4 + \frac{\rho^2 G^2}{2}}}$, $\eta_t = \frac{c}{(w + \sum_{i=1}^{t-2}G_i^2)^{1/3}}$ *with* $c \le \frac{2G^{2/3}}{\sqrt{K}}$ *and* $w = \max\{(4Lc)^3, 3G^2\}$, $\alpha_t = 2K\eta_t\eta_{t+1}$. *Algorithm 3 guarantees:*

$$\mathbb{E}\left[\frac{1}{T}\sum_{t=1}^{T}\|\nabla F(\vec{x}_t)\|\right] \le \frac{w^{1/6}\sqrt{2M} + 2M^{3/4}}{\sqrt{T}} + \frac{2\sigma_G^{1/3}}{T^{1/3}}$$

*Where* $M = \frac{1}{c}(20(\Delta + \frac{6\sigma_G^2 w^{1/3}}{5Kc}) + 192Kc^2\ln(T+1) + \frac{64G^4}{5K^2c^3}\ln T)$.

*Proof.* Define the potential:

$$\Phi_t = F(\vec{x}_t) + \frac{6}{5K\eta_t}\|\hat{\epsilon}_t\|^2$$

Then:
$$\mathbb{E}[\Phi_{t+1} - \Phi_t] = \mathbb{E}\left[F(\vec{x}_{t+1}) - F(\vec{x}_t) + \frac{6}{5K\eta_{t+1}}\|\hat{\epsilon}_{t+1}\|^2 - \frac{6}{5K\eta_t}\|\hat{\epsilon}_t\|^2\right]$$

Applying Lemma 10 and Lemma 2 then sum over t:

$$\mathbb{E}[\Phi_{T+1} - \Phi_1] \le \sum_{t=1}^T \mathbb{E}\left[-\frac{\eta_t}{4}\|\nabla F(\vec{x}_t)\|^2 + \frac{3\eta_t}{4}\|\hat{\epsilon}_t\|^2 + \frac{6}{5K\eta_{t+1}}\|\hat{\epsilon}_{t+1}\|^2 - \frac{6}{5K\eta_t}\|\hat{\epsilon}_t\|^2\right]$$

$$\le \mathbb{E}\left[\sum_{t=1}^T -\frac{\eta_t}{20}\|\nabla F(\vec{x}_t)\|^2 + \frac{48}{5}Kc^2\ln(T+1) + \frac{16G^4}{25K^2c^3}\ln T\right]$$

Reordering the term:

$$\mathbb{E}\left[\sum_{t=1}^T \eta_t\|\nabla F(\vec{x}_t)\|^2\right] \le \mathbb{E}\left[20(\Phi_1 - \Phi_{T+1}) + 192Kc^2\ln(T+1) + \frac{64G^4}{5K^2c^3}\ln T\right]$$

Also:

$$\mathbb{E}[\Phi_1 - \Phi_{T+1}] = \mathbb{E}[F(\vec{x}_1) - F(\vec{x}_{T+1}) + \frac{6}{5K\eta_1}\|\hat{\epsilon}_1\|^2 - \frac{6}{5K\eta_{T+1}}\|\hat{\epsilon}_{T+1}\|^2]$$

$$\le \Delta + \frac{6\sigma_G^2 w^{1/3}}{5Kc}$$

Plug in:

$$\mathbb{E}\left[\sum_{t=1}^T \eta_t\|\nabla F(\vec{x}_t)\|^2\right] \le 20\left(\Delta + \frac{6\sigma_G^2 w^{1/3}}{5Kc}\right) + 192Kc^2\ln(T+1) + \frac{64G^4}{5K^2c^3}\ln T$$

Now, let us relate $\mathbb{E}\left[\sum_{t=1}^T \eta_t\|\nabla F(\vec{x}_t)\|^2\right]$ to $\mathbb{E}\left[\sum_{t=1}^T \|\nabla F(\vec{x}_t)\|^2\right]$. Since $\eta_t$ is decreasing:

$$\mathbb{E}\left[\sum_{t=1}^T \eta_t\|\nabla F(\vec{x}_t)\|^2\right] \ge \mathbb{E}\left[\eta_T \sum_{t=1}^T \|\nabla F(\vec{x}_t)\|^2\right]$$

From Cauchy-Schwartz:

$$\mathbb{E}[1/\eta_T]\,\mathbb{E}\left[\eta_T \sum_{t=1}^T \|\nabla F(\vec{x}_t)\|^2\right] \ge \mathbb{E}\left[\sqrt{\sum_{t=1}^T \|\nabla F(\vec{x}_t)\|^2}\right]^2$$

Let $M = \frac{1}{c}\left(20(\Delta + \frac{6\sigma_G^2 w^{1/3}}{5Kc}) + 192Kc^2\ln(T+1) + \frac{64G^4}{5K^2c^3}\ln T\right)$, then:

$$\mathbb{E}\left[\sqrt{\sum_{t=1}^T \|\nabla F(\vec{x}_t)\|^2}\right]^2 \le \mathbb{E}\left[\frac{cM}{\eta_T}\right]$$

$$\le \mathbb{E}\left[M(w + \sum_{t=1}^T G_t^2)^{1/3}\right]$$

Let $\zeta_t = \nabla f(\vec{x}_t, z) - \nabla F(\vec{x}_t)$ so that $\mathbb{E}[\|\zeta_t\|^2] \le \sigma_G^2$. Then we have $G_t^2 = \|\nabla F(\vec{x}_t) + \zeta_t\|^2 \le 2\|\nabla F(\vec{x}_t)\|^2 + 2\|\zeta_t\|^2$. Thus:

$$\mathbb{E}\left[\sqrt{\sum_{t=1}^T \|\nabla F(\vec{x}_t)\|^2}\right]^2 \le \mathbb{E}\left[M\left(w + 2\sum_{t=1}^T \|\zeta_t\|^2\right)^{1/3} + 2^{1/3}M\left(\sum_{t=1}^T \|\nabla F(\vec{x}_t)\|^2\right)^{1/3}\right]$$

$$\le M(w + 2T\sigma_G^2)^{1/3} + \mathbb{E}\left[2^{1/3}M\left(\sqrt{\sum_{t=1}^T \|\nabla F(\vec{x}_t)\|^2}\right)^{2/3}\right]$$

$$\le M(w + 2T\sigma_G^2)^{1/3} + 2^{1/3}M\left(\mathbb{E}\left[\sqrt{\sum_{t=1}^T \|\nabla F(\vec{x}_t)\|^2}\right]\right)^{2/3}$$

Define $X = \sqrt{\sum_{t=1}^{T} \|\nabla F(\vec{x}_t)\|^2}$. The the above can be rewritten as:

$$(\mathbb{E}[X])^2 \leq M(w + 2T\sigma_G^2)^{1/3} + 2^{1/3}M(\mathbb{E}[X])^{2/3}$$

This implies that either $(\mathbb{E}[X])^2 \leq 2M(w+2T\sigma_G^2)^{1/3}$ or $(\mathbb{E}[X])^2 \leq 2 \times 2^{1/3}M(\mathbb{E}[X])^{2/3}$. Solving for $\mathbb{E}[X]$ in these two cases, we get:

$$\mathbb{E}[X] \leq \sqrt{2M}(w + 2T\sigma_G^2)^{1/6} + 2M^{3/4}$$

Finally, by Cauchy-Schwartz we have $\sum_{t=1}^{T} \|\nabla F(\vec{x}_t)\|/T \leq X/\sqrt{T}$. Therefore:

$$\mathbb{E}\left[\sum_{t=1}^{T} \frac{\|\nabla F(\vec{x}_t)\|}{T}\right] \leq \frac{w^{1/6}\sqrt{2M} + 2M^{3/4}}{\sqrt{T}} + \frac{2\sigma_G^{1/3}}{T^{1/3}}$$

with $M = \frac{1}{c}\left(20(\Delta + \frac{6\sigma_G^2 w^{1/3}}{5Kc}) + 192Kc^2 \ln(T+1) + \frac{64G^4}{5K^2c^3}\ln T\right)$ $\qquad\square$