# OpenReview forum: "Better SGD using Second-order Momentum"
_NeurIPS.cc/2022/Conference — NeurIPS 2022 Accept_

### Official Review · Reviewer_EDoL · 2022-07-08

**Rating:** 6
**Confidence:** 4
**Soundness:** 3 good
**Presentation:** 3 good
**Contribution:** 3 good

**Summary:**

This paper derives an improved momentum SGD method based on Hessian-vector products. It achieves optimal theorical rates on non-convex problems, and high-accuracy performance on deep learning tasks.

**Questions:**

The introduction says that the optimal rate of convergence is T=O(eps^-3), however, the rate of Theorem 1 seems to be faster than that (eps = logT/T^(2/3)), could you explain more on this optimality?

A discussion on the related work by Cutkosky,Mehta 2022 (Momentum Improves Normalized SGD) is needed I think in Section 3, which seems to be too short.

What is the practical performance of Algorithm 3? Some discussions are welcome. What is tilde O on line 196?

What is the batchsize used in the experiment (section 5.1 setup)? It is not clear. A numerical comparison or at least some more discussions with the method Cutkosky et al. 2020  (Momentum-based variance reduction in non-convex sgd) is needed in Section 5 to support the claim in the introduction that recursive variance reduction requires large batches, as numerical results in that paper seem to work well on CIFAR.

Why the top 1 test accuracy, bleu score are higher with SGDHess compared to SGD in Table 1 and 2? Is that due to a higher training accuracy or some form of implicit regulization?



**Ethics Review Area:**

["I don’t know"]

**Strengths And Weaknesses:**

The strength of this paper is to introduce a correction to reduce the bias in the classical momentum SGD method (section 2). An algorithm SGDHess is proposed for this purpose, with theorical convergence analysis. Although there is a gap between the theorical version of the algorithm, and its practical version (i.e. return hat x in theory, and x_T in practice), numerical results are convincing about its improved accuracy compared to momentum SGD. One weakness is that I do not fully understand why it works better in practice, could this be explained better (see questions below)?

---

> ### Author Response · Authors · 2022-08-02
> **Response to reviewer EDoL**
>
> Thank you for your comments! Your comments are very helpful for us to improve the quality of our paper. We will address some of your concerns below.
>
> Response to questions:
>
> 1. Sorry for the confusion! The guarantee of the first algorithm is slightly different since it is the bound on the sum of gradient squares, which is O(1/T^{⅔}). If we convert to a bound on the gradient norm itself, we obtain the optimal rate(1/T^{⅓}). We will modify this in the final revision.
>
> 2. Thanks! We will add a discussion in the appendix.
>
> 3. The $\tilde O$ means we hide logarithmic factors in T.
>
> 4. We run with batch size=256. All the batch sizes used are either recommended by [1] or commonly used in previous experiments. For more details, please refer to the appendix.
>
> 5. We suspect that just by dint of being a better optimization method (in theory), sgdhess is able to achieve better training errors, which gives it a better bleu score. This is also consistent with our observation from image classification tasks where sgdhess also consistently achieve smaller training error than sgd.

---

### Official Review · Reviewer_nnCX · 2022-07-10

**Rating:** 7
**Confidence:** 4
**Soundness:** 4 excellent
**Presentation:** 4 excellent
**Contribution:** 4 excellent

**Summary:**

The paper introduces a Hessian-based correction term in SGD with momentum to reach $\epsilon$ critical point in $\mathcal{O}(\epsilon^{-3})$ stochastic gradient and Hessian-vector product computations. The authors extend the benefit of the correction term to normalized SGD and adaptive algorithms. Finally, they showcase the benefits of their proposed algorithm with an experimental study on CIFAR-10, ImageNet, and neural machine translation.

**Questions:**

Here are my major questions:
1. It would be great to have a discussion on why Arjevani et al. [2021] require large batch sizes and how the authors can tackle the necessity with momentum. The algorithm of Arjevani et al. [2021] (algorithm 1) uses K Hessian vector products (similar to the one being used in the current paper) per gradient step to reduce the variance. However, the algorithm of Arjevani et al. uses different samples for the K Hessian vector products, while the authors can get away with a single Hessian vector product update in the momentum. Why?


2. There seems to be a mismatch between the theoretical claims and the experimental results. The theoretical claims show that SGDHess can reach a stationary point with $\mathcal{O}(\epsilon^{-3})$  stochastic gradient and Hessian calls, compared to $\mathcal{O}(\epsilon^{-4})$ stochastic calls for SGD. However, the experiments talk about achieving better generalization with SGDHess.

a) How is the result connected to generalization?

b) Isn't it better to compare the loss behavior for the two algorithms, and show faster convergence for SGDHess?

3. ResNet models use the ReLU activation function and maximum pooling layers. Hence, the global smoothness and the second-order smoothness are undefined, which are necessary for the theoretical claims. One suggestion to modify the experiments will be to use smooth activation functions like GeLU and tanh, and average pooling layers.



**Limitations:**

The authors have jotted down one of the main limitations of their work, which is assuming that the function is Lipschitz. However, it would be great to also include a discussion on the viability of the other assumptions, which are L-smoothness (Eq. 2), second-order smoothness (Eq. 6), and the upper bound on the variance of the gradient and the Hessian (Eqs. 3 and 4). Since one of the major ambitions of the paper is to show the benefits of the proposed algorithm in neural networks, it would be great to see some ablations for these assumptions in the deep learning architectures.

**Strengths And Weaknesses:**

The major strength of the work is in the simplicity of the correction term introduced in momentum, which achieves the same complexity to reach an $\epsilon$-critical point, compared to Arjevani et al. [2021], which requires a larger batch size (and a more complicated Hessian vector product calls). Moreover, the authors can show better performance of SGDHess compared to SGD in various vision and NLP settings. The theorem statements are clearly written and easy to understand.


The major issue with this is that the global smoothness and second-order smoothness constants may be very large for the theoretical bounds to turn vacuous for the deep neural networks. Moreover, there is a mismatch between the theoretical claims and the experimental results as pointed out later.

---

> ### Author Response · Authors · 2022-08-02
> **Response to reviewer Reviewer nnCX**
>
> Thank you for your comments! Your comments are very helpful for us to improve the quality of our paper. We will address some of your concerns below.
>
> Response to questions:
>
> 1 It is quite hard to point out the exact point in the analysis that allows us to dispense the batch size assumption but we suspect that due to our careful analysis and the recursive nature of momentum update, we are able to keep the biasedness small while [1] needs additional batch size to control this error.
>
> 2. This is a common issue in optimization analysis: we relax the true objective (generalization accuracy) several times: first by applying a loss function, then by assuming i.i.d. Inputs (e.g. by moving to training loss), and finally by looking for stationary points rather than true minimizers. Tightening this standard chain of relaxation is a significant research undertaking in and of itself. By dint of our improved convergence rates for this relaxation, we would hope for better generalization error, but you are certainly correct that for multiple passes over the dataset our results technically apply to the training error. The reason we include the testing error is that we feel at the end of the day, the testing error is the most important metric in practical tasks. Our goal is to show that it’s possible to design a second-order algorithm that enjoys both good theoretical and practical guarantees.
>
> 3. Thanks for the suggestion!  Relu networks are indeed not smooth by themselves, but the population loss may be more smooth if the distribution of input examples is smooth (as one might expect images to be, especially given natural smoothing effects like noise in the camera’s sensors) - consider that the expected value of a relu under Gaussian inputs is actually smooth. Thus, (perhaps surprisingly) the non-smoothness ReLU actually does not immediately disqualify our smoothness assumptions. This is one advantage of our algorithm over variance-reduction algorithms which require that each $f(x_t,z_t)$ is smooth. Nevertheless, we freely admit that our model may not perfectly match all neural networks in practice. This is in fact a problem with almost all theoretical analyses of optimization algorithms typically used in deep learning today. It is plausible that more refined analyses using smoothed versions of the objectives may alleviate this issue, but we feel this is out of scope for a single paper.
>
> [1] Arjevani, Yossi, et al. "Second-order information in non-convex stochastic optimization: Power and limitations." Conference on Learning Theory. PMLR, 2020.

---

> > ### Comment · Reviewer_nnCX · 2022-08-08
> > **Response after rebuttal**
> >
> > Thank you for the rebuttal. I would ask the authors to think a bit more about 1. When I was reading your proofs (and proof sketch), I felt I was reading a very conventional optimization proof. This didn't give a sense of what "barrier" you have to jump to go from the result of [1].
> > In particular, why should a "second-order momentum" help reduce the bias in the updates?
> >
> >
> > [1] Arjevani, Yossi, et al. "Second-order information in non-convex stochastic optimization: Power and limitations." Conference on Learning Theory. PMLR, 2020.

---

> > > ### Author Response · Authors · 2022-08-09
> > > **Response to reviewer nnCX**
> > >
> > > We thank the reviewer for the question!
> > >
> > > We think the main problem here is that Algorithm 1 in [1] just inherently doesn’t work without a large batch size. Algorithm 1 in [1]  is based on SARAH [2] which requires the optimizer to infrequently compute a checkpoint (which is also used in SVRG). Algorithm 1 of [1] computes the checkpoint gradient using a large batch size roughly every $O(1/\epsilon)$ iterations. The convergence of SARAH depends on the error term $||\nabla F(w) - g_t||^2$ (analysis of Theorem 2 in [2]) where $g_t$ is the current gradient estimate of SARAH. This error in turn depends on the variance of the checkpoint gradient (analysis of Lemma 3 in [2]). Thus without the batch size, the variance of this checkpoint would be large (since now the variance is $O(1)$ instead of $O(1/B)$), which causes issues in the convergence analysis. On the other hand, in our analysis, even though our error also depends on the term $||\nabla F(w) - g_t||^2$ (equation below equation 15 - page 20), we have the momentum parameter to control this error. Intuitively, we can think of the second-order momentum as providing us with an extra knob that we can use to control the error, which leads to a good convergence rate without the help of large batch size.
> > >
> > >
> > > [1] Arjevani, Yossi, et al. "Second-order information in non-convex stochastic optimization: Power and limitations." Conference on Learning Theory. PMLR, 2020.
> > >
> > > [2] Nguyen, Lam M., et al. "SARAH: A novel method for machine learning problems using stochastic recursive gradient." International Conference on Machine Learning. PMLR, 2017.

---

### Official Review · Reviewer_mZiQ · 2022-07-12

**Rating:** 6
**Confidence:** 4
**Soundness:** 3 good
**Presentation:** 3 good
**Contribution:** 3 good

**Summary:**

The paper proposed a supposed novel way of using the Hessian-vector product as a momentum vector instead of the traditional momentum vector. The gradient is computed as a linear combination of the past gradient, which is also supposedly clipped, and the current gradient.

**Questions:**

1. Why is $z_t$ appearing everywhere ? If the gradients are not computed with respect to $z_t$, it is recommended to remove it.
2. What is G-Lipschitz ? The formulation in equation (6) is very different from the one the reviewer has seen. Can the authors share a reference ?
3. What is L ? Is this the constant term of the Lipschitz continuity equation ? If so, how is it determined ?
Editorial:
4. Why are only equations 1-6 numbered ?

**Limitations:**

1. The paper does not identify any convergence criterias.
2. It is not performing better than state-of-the-art. It might seem in Figure 1(a) and (b) that they might reach maximum accuracy, but the sharp increase is also indicative of overtraining
3. Equation with the gradient update (please use number for the main equations) can literally be boiled down to a linear combination of the past gradient and the current gradient.  $\hat{g}_t = (1-\alpha)\hat{g}_{t-1} + \alpha g_t$

**Strengths And Weaknesses:**

Strengths:
1. Used the Hessian vector product is computed efficiently.
Weaknesses:
1. There is no convergence guarantee for the current method
2. The computation expense is higher for the proposed approach, but the improvement in accuracy is limited. Especially in the case of Imagenet.
3. The authors mention that the level of noise is limited in the conclusion section. However, no evidence of this is provided.
4. Upon closer inspection in the appendix material, the results do no show improvement to

Editorial:
1. The paper is very hard to read. A lot of the derivations and lemmas can be moved to the supplementary section.
2. The presentation needs improvement. Give more importance to the convergence proofs and the results. All of the derivations seem to reduce the credibility of the paper.

---

> ### Author Response · Authors · 2022-08-02
> **Response to reviewer mZiQ**
>
> Thank you for your comments! Your comments are very helpful for us to improve the quality of our paper. We will address some of your concerns below.
>
> Response to questions:
>
> 1.Since $z_t$ is the data that we sample from the data set, the gradient at any particular time $t$ does depend on $z_t$. Thus, we think writing the gradient as $\nabla f(x_t,z_t)$ is necessary since it depends on both the sampled data and the current iterate.
>
> 2. The definition you may be familiar with is that a function $f$ is G-Lipschitz w.r.t \|.\| if for any $x,y$, $\|f(x)-f(y)\| \le G\|x-y\|$. For differentiable $f$, this is equivalent to $\|\nabla f(x)\|_\star \le G$ for every x. This is a standard fact covered in optimization and analysis classes and is for example the first “property” on the Wikipedia page for Lipschitz continuity.
>
> 3. As stated in line 95, L is the smoothness constant. This is a very common assumption in non-convex optimization since it’s very hard to prove any convergence rate for non-smooth, non-convex function (Some other papers that use this assumption: [8], [9],[10]).
>
> 4. We number assumptions 1-6 so that it’s easier for the reader to see which algorithm requires which assumption. We are happy to number other equations as well.
>
> Response to limitation:
> 1. Sorry for any confusion, we will add a sentence to discuss this. Our convergence criterion is the upper bound on the expected norm of the gradients. We want to stress that this is a common convergence criterion in non-convex optimization ([5],[6])
>
> 2. The main result of our paper is in the theoretical front when we achieve the optimal convergence rate of $O(1/T^{1/3})$  while even the fastest first-order algorithm can only achieve $O(1/T^{1/4})$ rate ([1] has shown that no first-order algorithm can exceed the $O(1/T^{1/4})$ rate without using any second-order information or variance-reduction techniques). And even when we extend the comparison to other second-order algorithms, our algorithm still has advantages over the literature. Most second-order algorithms either are unable to achieve the optimal  $O(1/T^{1/3})$ rate ([2], [3]) or do not have convergence guarantees at all ([4]). The only second-order algorithm that we’re aware of that also achieves the optimal rate is algorithm 1 in [5]]. However, their algorithm requires the batch size to be roughly $O(T^{1 / 3})$, which is excessively large for practical use-cases.  On the experiment front, we show a slight improvement in image classification tasks and significantly bridge the gap between SGD and Adam in the language modeling task.
>
> 3. We’re not sure what you mean by this. Our main update is $\hat g_t = (1-\alpha)(\hat g_{t-1} + \nabla^2 f(x_t) (x_t - x_{t-1}) + \alpha g_t$. If we take out the hessian-vector product part, the algorithm will recover exactly the update of SGD with momentum, which is known to have O(1/T^{¼}) convergence rate instead of the O(1/T^{⅓}) that we achieve.
>
> Citations:
>
> [1] Arjevani, Yossi, et al. "Lower bounds for non-convex stochastic optimization." arXiv preprint arXiv:1912.02365 (2019).
>
> [2] Tripuraneni, Nilesh, et al. "Stochastic cubic regularization for fast nonconvex optimization." arXiv preprint arXiv:1711.02838 (2017).
>
> [3] Zhou, Dongruo, Pan Xu, and Quanquan Gu. "Stochastic variance-reduced cubic regularized Newton methods." International Conference on Machine Learning. PMLR, 2018.
>
> [4] Yao, Zhewei, et al. "ADAHESSIAN: An adaptive second order optimizer for machine learning." arXiv preprint arXiv:2006.00719 (2020).
>
> [5] Arjevani, Yossi, et al. "Second-order information in non-convex stochastic optimization: Power and limitations." Conference on Learning Theory. PMLR, 2020.
>
> [6] Cutkosky, Ashok, and Francesco Orabona. "Momentum-based variance reduction in non-convex sgd." Advances in neural information processing systems 32 (2019).
>
> [7] Bubeck, Sébastien. "Convex optimization: Algorithms and complexity." Foundations and Trends® in Machine Learning 8.3-4 (2015): 231-357.
>
> [8] Fang, Cong, et al. "Spider: Near-optimal non-convex optimization via stochastic path-integrated differential estimator." Advances in Neural Information Processing Systems 31 (2018).
>
> [9] Ghadimi, Saeed, and Guanghui Lan. "Stochastic first-and zeroth-order methods for nonconvex stochastic programming." SIAM Journal on Optimization 23.4 (2013): 2341-2368.
>
> [10] Johnson, Rie, Tong Zhang. ‘Accelerating Stochastic Gradient Descent using Predictive Variance Reduction’. Advances in Neural Information Processing Systems. Επιμέλ. C. J. Burges κ.ά. τ. 26. Curran Associates, Inc., 2013. Web.

---

> ### Comment · Reviewer_mZiQ · 2022-08-07
> **Response to authors**
>
> The reviewer thanks the authors for answering all the concerns.
>
> I believe the authors have done a good job of explaining all the issues. This significantly improves the rating of the paper.
>
> However, the presentation can be slightly improved. Thus, I will place my rating accordingly.

---

### Meta-Review · Area_Chair_ipBj · 2022-08-26

**Recommendation:** Accept
**Confidence:** Certain

**Metareview:**

This papers develops a new efficient second order method for non-convex optimization. Reviewers seemed to agree that the paper offers a significant over the prior work of Arjevani, Yossi, et al "Second-order information in non-convex stochastic optimization: Power and limitations" by requiring a single Hessian vector product update in the momentum update, and therefore would be of interest to the NeurIPS audience.

**Award:**

No

---

### Decision · Program_Chairs · 2022-09-14

Accept